# CINETRANS: LEARNING TO GENERATE VIDEOS WITH CINEMATIC TRANSITIONS VIA MASKED DIFFUSION MODELS

**Xiaoxue Wu**[1,2]   **Bingjie Gao**[2,3]   **Yu Qiao**[2†]   **Yaohui Wang**[2†]   **Xinyuan Chen**[2†]
[1]Fudan University , [2]Shanghai Artificial Intelligence Laboratory , [3]Shanghai Jiao Tong University

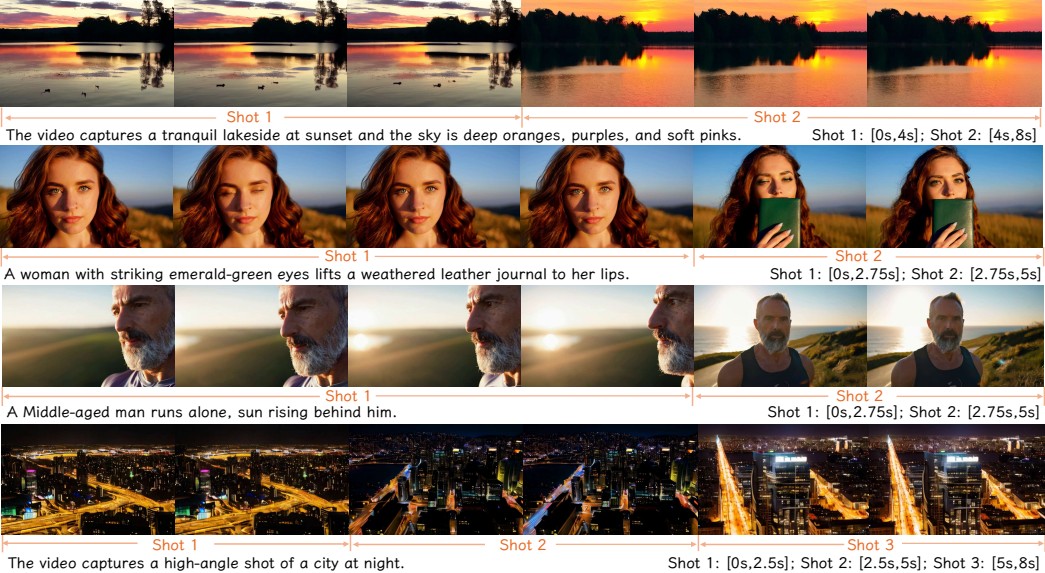

Figure 1: Multi-shot videos generated by CineTrans, which enables cinematic transitions aligning with film editing. The corresponding mask is constructed based on the timestamps of the shots, thereby controlling cinematic transitions Project page: *https://uknowsth.github.io/CineTrans/*

## ABSTRACT

Despite significant advances in video synthesis, research into multi-shot video generation remains in its infancy. Even with scaled-up models and massive datasets, the shot transition capabilities remain rudimentary and unstable, largely confining generated videos to single-shot sequences. In this work, we introduce **CineTrans**, a novel framework for generating coherent multi-shot videos with cinematic, film-style transitions. To facilitate insights into the film editing style, we construct a multi-shot video-text dataset **Cine250K** with detailed shot annotations. Furthermore, our analysis of existing video diffusion models uncovers a correspondence between attention maps in the diffusion model and shot boundaries, which we leverage to design a mask-based control mechanism that enables transitions at arbitrary positions and transfers effectively in a training-free setting. After fine-tuning on our dataset with the mask mechanism, CineTrans produces cinematic multi-shot sequences while adhering to the film editing style, avoiding unstable transitions or naive concatenations. Finally, we propose specialized evaluation metrics for transition control, temporal consistency and overall quality, and demonstrate through extensive experiments that CineTrans significantly outperforms existing baselines across all criteria.

---

[†]Corresponding authors.

# 1 INTRODUCTION

Endowed with extensive pre-training and sophisticated architectures, diffusion models (Blattmann et al., 2023; Ho et al., 2020; Rombach et al., 2022; Song et al., b) have demonstrated promising capabilities in generating videos with high visual quality and strong consistency comparable to real videos (Wan et al., 2025; Kong et al., 2024; Yang et al., 2024; Li et al., 2024). Particularly in the domain of text-to-video generation (T2V) (Wang et al., 2024b; Yang et al., 2024; Li et al., 2023; Cho et al., 2024), the breakthrough has attracted considerable attention, highlighting the potential of diffusion models in mastering video creation (Wang et al., 2024b; Xu et al., 2024; Chen et al., 2023a; 2024a). However, generating multi-shot videos with cinematic editing style and movie-like transitions from a brief user input continues to be a significant challenge.

Existing work on long video generation can be broadly categorized into two aspects. First, larger models trained on massive datasets advance the capability of video interpretation and generation, enhancing the visual fidelity and maximum video length. (Kong et al., 2024; Wan et al., 2025; Yang et al., 2024). Second, techniques such as conditioning improve consistency across concatenated samples, enabling longer outputs through clip stitching (Zheng et al., 2024; He et al., 2023; Xie et al., 2024; Zhao et al., 2024). While both approaches partially support multi-shot sequence generation, they still exhibit notable limitations. Large-scale models primarily focus on single-shot videos due to the scarcity of shot transitions in their training datasets (Wang et al., 2023b; Ju et al., 2025; Wang et al., 2023a). Cinematic transitions are not guaranteed, let alone at precisely controlled positions. Moreover, the high computational cost and extended training time undermine the efficiency of this approach. Meanwhile, generating individual shots separately and concatenating them requires substantial manual intervention, and ignores prior knowledge from cinematic multi-shot dataset, resulting in cuts that often misalign with real-world editing styles. Additionally, many recent works often target narrow contexts, such as facial consistency (Zheng et al., 2024) or specific animated series (Dalal et al., 2025), which constrains their applicability to general video generation.

While the aforementioned methods achieve certain multi-shot effects, very few explicitly focus on cinematic transitions within diffusion models. In this work, we propose **CineTrans**, a framework that produces multi-shot videos with cinematic transitions as shown in Figure 1. To support this, we develop a refined data processing pipeline that processes raw footage into a dataset of 250K video-text pairs. Our split-stitch procedure groups semantically related clips and removes gradual transitions, and we employ transition-aware models to annotate hierarchical captions. The resulting **Cine250K** provides frame-level shot labels and temporally-dense captions, preserving film editing style and making it well-suited for multi-shot video generation, while proving effective in enabling more natural shot transitions and stronger inter-shot consistency.

To analyze how diffusion-based models handle cinematic multi-shot sequences, we dive deep into the attention patterns, examining attention maps across the temporal dimension. We find that the attention maps have strong impact on the intra-shot and inter-shot frames. This insight clarifies the underlying mechanism of shot transitions in diffusion models. Building on this, we introduce a mask mechanism featuring strong correlations within shots and weak correlations between shots, achieving controlled cinematic transitions and enabling zero-shot multi-shot generation.

As shown in Figure 2, our proposed CineTrans framework functions through two key aspects. First, through an analysis of the attention maps, we prove that the mask mechanism aligns with the diffusion model's inherent understanding of cinematic multi-shot sequences. The application of mask enables strong intra-shot frame correlations in attention module, facilitating precise frame-level cinematic transitions, which remains effective even in a training-free setting. Second, the constructed Cine250K encapsulates prior knowledge of film editing. Fine-tuning on this dataset equips Cine-Trans with the ability to generate cinematic transitions that conform to this style, rather than simply concatenating semantically similar clips. Consequently, CineTrans is able to producing cinematic multi-shot videos aligned with film-editing conventions.

We evaluate the model on a series of prompts with specified transitions and conduct a comprehensive analysis using multiple metrics from different perspectives. Experimental results demonstrate that CineTrans achieves finer shot transition control and stronger consistency compared to other multi-shot generation methods, without compromising overall quality.

Our contributions are summarized as follows:

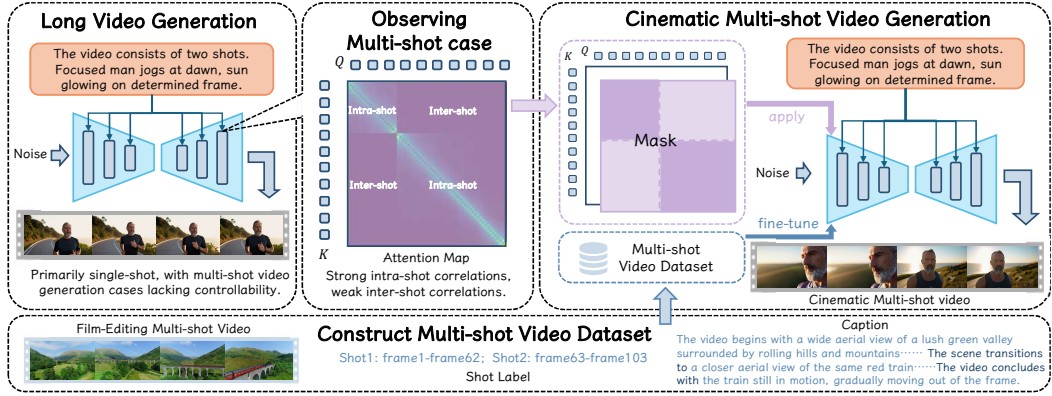

Figure 2: **Overview of CineTrans.** Existing video generation models focus primarily on single-shot video. The multi-shot video generation cases often follow several failures, remaining unstable and uncontrollable. Observations of these multi-shot cases reveal a structured pattern in attention layers. Based on this insight, we introduce a mask mechanism and fine-tune the model with our constructed dataset Cine250K, resulting in significantly improved performance.

- We developed a dataset of 250K video-text pairs, complete with frame-level shot labels and hierarchical annotations, which facilitates video diffusion models for generating cinematic transitions and consistency between shots.
- We analyze attention maps in diffusion models for multi-shot video generation and observe a strong connection between attention probabilities and shot transitions. Building on this insight, we introduce a mask mechanism that enables cinematic transitions within diffusion models, leading to the CineTrans framework, which is effective in a training-free setting.
- We propose a series of comprehensive metrics tailored for cinematic multi-shot video generation and evaluate CineTrans, demonstrating its ability to control cinematic transitions, enhance temporal consistency, and preserve overall quality.

## 2 RELATED WORK

**Diffusion-based Video Generation.** Diffusion-based approaches, built on the iterative denoising framework of Latent Diffusion Model (Rombach et al., 2022), utilize scaled-up model (Kong et al., 2024; Wan et al., 2025; Yang et al., 2024; Blattmann et al., 2023; Chen et al., 2023a; 2024a; He et al., 2022; Lin et al., 2024; Polyak et al., 2024) and large video datasets (Miech et al., 2019; Bain et al., 2021; Zellers et al., 2021; Xue et al., 2022; Wang et al., 2023a;b; Chen et al., 2024b; Xiong et al., 2025) to generate high-quality, prompt-adherent videos with extended durations. Owing to strong semantic capacity, certain pretrained models (Kong et al., 2024; Wan et al., 2025) can preliminarily generate multi-shot videos when provided with transition-specified prompts, but require vast computational resources, long training, and yield imprecise transitions. In contrast, our work offers frame-level stable cinematic transitions, seamlessly applied to the diffusion-based framework.

**Multi-Shot Video Generation.** Recent work has explored multi-shot video generation, which can be categorized into two main approaches. The first generates each shot separately and then concatenates them, focusing on consistency between the generated shots. Animate-a-Story (He et al., 2023) uses motion structure retrieval for plot-aligned clips guided by text prompts. SEINE (Chen et al., 2023b) proposes a mask-based diffusion model to generate a smooth transition between shots. Vlogger (Zhuang et al., 2024) attempts to utilize the language model to generate prompts for different shots of a video. DreamFactory (Xie et al., 2024) employs an LLM-based framework with multi-agent collaboration and keyframe iteration design. MovieDreamer (Zhao et al., 2024) adopts a hierarchical autoregressive architecture for global coherence, and VGoT (Zheng et al., 2024) uses keyframes and identity-preserving embeddings to enforce temporal and character consistency. While these consistency-driven approaches are effective to some extent, they do not leverage real multi-shot video datasets and tend to overlook complex relationships between shots, including variations

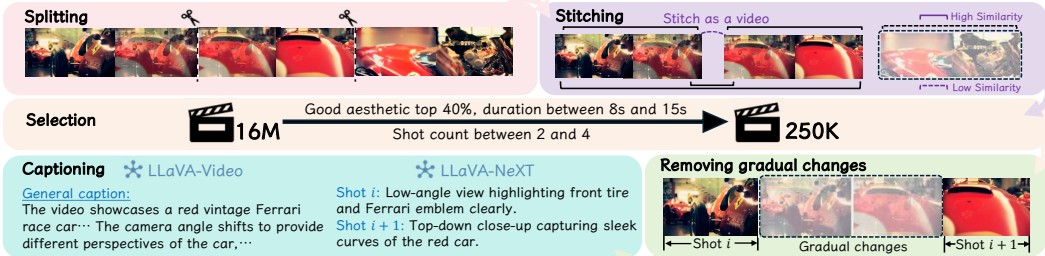

Figure 3: **Dataset curation pipeline.** The raw video is split into several clips and then selectively stitched based on semantic features. A selection process then chooses high-quality multi-shot videos. After initial assembly, gradual changes are removed. Finally, a language model is used to annotate each video with a general caption and each shot with its shot caption, yielding temporally dense annotations.

in camera angles and scenes, which are crucial in human-edited videos. The second approach modifies model architectures to generate multi-shot content directly. Test-Time training (Dalal et al., 2025) adds a layer for long multi-shot generation but is tied to a specific animated series, limiting generalization. Mask$^2$DiT (Qi et al., 2025) applies shot-wise semantic masking yet focuses on text injection and assumes fixed shot durations. ShotAdapter (Kara et al., 2025) introduces transition tokens but exhibits low consistency. LCT (Guo et al., 2025) enhances multi-shot understanding with specialized positional encodings but demands large-scale training. In contrast, our method generates cinematic multi-shot videos in a single pass with flexible frame-level control, achieving strong performance even in a training-free setting and demonstrating generalizability across diverse scenarios.

**Temporal-Controlled Video Generation.** Several recent works in video generation focus on controlling the temporal dimension, enabling the synthesis of longer videos with more dynamic and fine-grained temporal control Cai et al. (2025); Li et al. (2025); Ouyang et al. (2024). VSTAR Li et al. (2025) draws inspiration from the band-matrix-like structure observed in the attention maps of real videos, and introduces a Temporal Attention Regularization strategy to enhance temporal dynamics and motion continuity during generation. Our targeted task, multi-shot video generation, can be regarded as a variant of temporally controlled video generation. It shifts the temporal control from frame-level semantic evolution to shot-level semantic transitions, which better aligns with real-world video editing practices and the narrative flow of long-form videos. Fundamentally, our proposed mask mechanism, like other temporally controlled generation methods, manipulates the attention distribution among visual tokens within the diffusion process to achieve more coherent and controllable video synthesis.

**Masked Attention Mechanism.** Masked attention has been widely used to regulate contextual dependencies and enable controllable generation. It has been applied in representation learning Bao et al. (2022); Tong et al. (2022), generative modeling Chang et al. (2022); Yu et al. (2023); Luo et al. (2023), and fine-grained spatial or compositional control in diffusion models Endo (2024); Wang et al. (2024a). Our work follows this line but employs attention masking for shot-level temporal control in video diffusion, ensuring coherence across shots while preserving intra-shot consistency.

## 3 DATASET

A video with cinematic transitions integrates multiple clips while preserving consistency. To capture film editing prior knowledge for multi-shot sequences, we introduce Cine250K, a dataset for cinematic video generation. As shown in Figure 3, starting from 633K richly edited videos from Vimeo[1], we design a multi-stage preprocessing pipeline to construct the annotated multi-shot dataset.

First, the transition points are identified by Pyscenedetect (Castellano, 2024), resulting in fragmented segments. Adjacent segments with high similarity, measured by ImageBind (Girdhar et al., 2023) features, are then stitched together according to predefined rules to assemble an initial collection of 16M clips containing shot transitions. We filter this collection by aesthetic score, duration, and

---

[1]https://vimeo.com

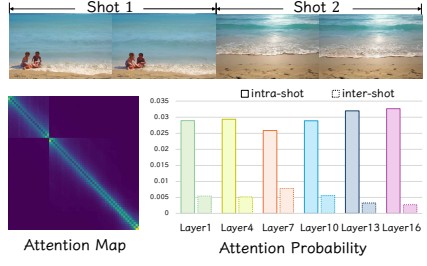

(a) A case of generated multi-shot video within diffusion model.

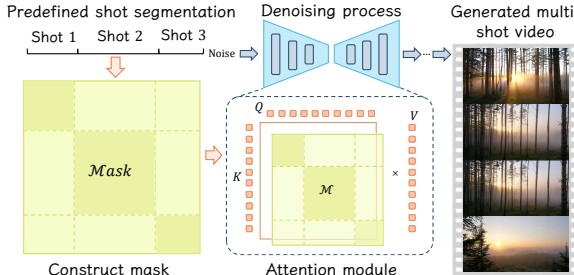

(b) The architecture of the mask mechanism.

Figure 4: We observe that in multi-shot scenarios the attention maps form a block-diagonal pattern, i.e., certain layers exhibit higher intra-shot than inter-shot frame correlations, so we design a corresponding masking mechanism. Using predefined transition points, the mask is applied to those layers of the diffusion model to guide cinematic multi-shot video generation.

shot count to form the preliminary video dataset. Subsequently, since shot transitions can be categorized into instantaneous hard cuts and gradual changes that blur segment boundaries, we apply TransNetV2 (Soucek & Lokoc, 2024) to detect and remove all gradual transition frames which do not clearly belong to any single shot. This yields unambiguous segments with precise shot labels, including exact start and end frame indices. In the final step, each video receives both a general caption produced by LLaVA-Video-7b-Qwen2 (Zhang et al., 2024) for temporally-dense descriptions of cinematic transitions, and separate shot captions from LLaVA-NeXT (Liu et al., 2024).

Following the pipeline, Cine250K offers high-aesthetic videos, precise shot labels, and hierarchical captions, thereby supplying rich prior knowledge for cinematic multi-shot video generation and facilitating the production of videos with authentic film editing style. The details and comparison with previous video datasets are presented in Appendix A and B.

## 4 METHODOLOGY

In this section, we introduce our insight and method for the proposed CineTrans. We first provide the necessary preliminaries in Section 4.1 to facilitate the following discussions. Section 4.2 presents our observations in the attention maps of diffusion models, highlighting the differences between intra-shot and inter-shot correlations. Based on this observation, we introduce a block-diagonal mask mechanism detailedly in Section 4.3 for cinematic multi-shot video generation. In Section 4.4, we discuss the implementation of inference. An overview of the methodology is shown in Figure 4.

### 4.1 PRELIMINARY

Diffusion models (Ho et al., 2020; Song et al., a) are a class of generative models learning to reverse a diffusion process, which learn to corrupt data via a predefined noisy Gaussian process and then invert that corruption through a trained neural network. Video diffusion models simply apply this same forward-reverse framework across temporal frames, yielding coherent video sequences $\mathbf{F} = \{f_1, f_2, \ldots, f_N\}$, where each element $f_t$ represents a video frame. A key component in these models is integrating the attention mechanism (Vaswani et al., 2017), which allows latent variables to focus on each other's relevant information and can be formalized as:

$$\text{Attention}(\mathbf{Q}, \mathbf{K}, \mathbf{V}) = \text{softmax}\left(\frac{\mathbf{Q}\mathbf{K}^T}{\sqrt{d_k}}\right)\mathbf{V}, \tag{1}$$

where $\mathbf{Q}$, $\mathbf{K}$, $\mathbf{V}$ are the query, key, and value matrices, and $d_k$ is the key dimensionality.

Cinematic multi-shot video generation task also follows the framework mentioned above, aiming to generate videos with cinematic transitions that are both aesthetically pleasing and compliant with the text prompts. Additionally, with the introduction of multi-shot, the core challenges include:

- **Generation of cinematic transitions**. The generated sequence $\mathbf{F} = \{f_1, f_2, \ldots, f_N\}$ can be divided into $M$ sub-intervals, where each $\mathbf{F}_m = \{f_{i_m}, f_{i_m+1}, \ldots, f_{i_{m+1}-1}\}$ contains frames from index $i_m$ to $i_{m+1} - 1$, for $m = 1, 2, \ldots, M$, $M$ is the specified shot count, and $i_1 = 1, i_{M+1} = N + 1$. Within sub-intervals, frames maintain visual continuity, while noticeable changes at the boundaries create cinematic transitions.
- **Consistency within and across shots**. Within a shot, visual consistency is crucial for continuity. In contrast, inter-shot consistency emphasizes high-level semantic similarity over low-level details, i.e., maintaining consistency across shots despite substantial compositional differences. This means that its evaluation is not confined to specific attributes, such as composition or facial features, but is guided by film-editing conventions, ensuring applicability to general scenarios.

## 4.2 FRAME CORRELATION IN ATTENTION MODULE

Recently, owing to the increased model size, dataset scale, and computational resources, some diffusion models (Kong et al., 2024; Wan et al., 2025) have demonstrated preliminary capabilities in generating multi-shot videos. However, how diffusion models internally model transitions within a video remains unclear and warrants further exploration. We hypothesize that the correlation between adjacent frames at transition points is significantly different from that at non-transition points. The transition point involves a substantial shift, while a non-transition point requires continuity to maintain visual coherence, making them inherently divergent.

In video diffusion models, the denoising process captures temporal correlations through the attention module. Specifically, the video latent representation is flattened into a sequence of tokens. Tokens from different frames participate in the calculation of the attention maps in specific layers, enabling the pretrained model to generate continuous, high-quality video segments. As a result, attention maps are a valuable tool for analyzing frame correlations.

We explore and visualize frame-wise attention maps in the context of the multi-shot video generation cases. As shown in Figure 4a, it demonstrates strong correlations for intra-shot frames and weak correlations for inter-shot frames. More specifically, the attention probability matrix exhibits a block-diagonal structure, with each block corresponding to a shot. Figure 4a also illustrates the quantitative differences in attention probabilities across various layers, highlighting the variations between intra-shot and inter-shot correlations. To quantify this observation, we compute the average ratio of mean intra-shot to inter-shot attention probabilities (26.68) and assess its correspondence with the ground-truth shot boundaries via Pearson correlation, yielding r=0.71 (p<0.01). This suggests the potential of leveraging the attention maps to guide the generation of multi-shot videos.

## 4.3 MASK MECHANISM

Building on our observation, we introduce a mask mechanism, a simple strategy that operates on the attention probability in text-to-video diffusion models. Specifically, we construct an attention mask $\mathcal{M}$ for the visual tokens in the attention module at specific layers as follows:

$$\mathcal{M}_{ij} = \begin{cases} 0 & \text{if } i, j \in \text{same shot} \\ -\infty & \text{if } i, j \notin \text{same shot} \end{cases} \tag{2}$$

The mask matrix is subsequently added to the attention score in Eq. 1, effectively weakening the correlations across different shots:

$$\text{Attention}(\mathbf{Q}, \mathbf{K}, \mathbf{V}) = \text{softmax}\left(\frac{\mathbf{Q}\mathbf{K}^T}{\sqrt{d_k}} + \mathcal{M}\right)\mathbf{V}. \tag{3}$$

As a result, the final attention probabilities form block-diagonal matrices, with cinematic transitions occurring at predefined positions. As shown in Figure 4b, transition positions are specified in advance, and the mask matrix is constructed accordingly, enabling precise control over the process. With the block-diagonal mask mechanism, diffusion models can generate cinematic multi-shot videos with fine-grained control.

The mask mechanism functions through two aspects. First, it aligns with the phenomenon observed in Section 4.2, conforming to diffusion models' inherent understanding of cinematic multi-shot

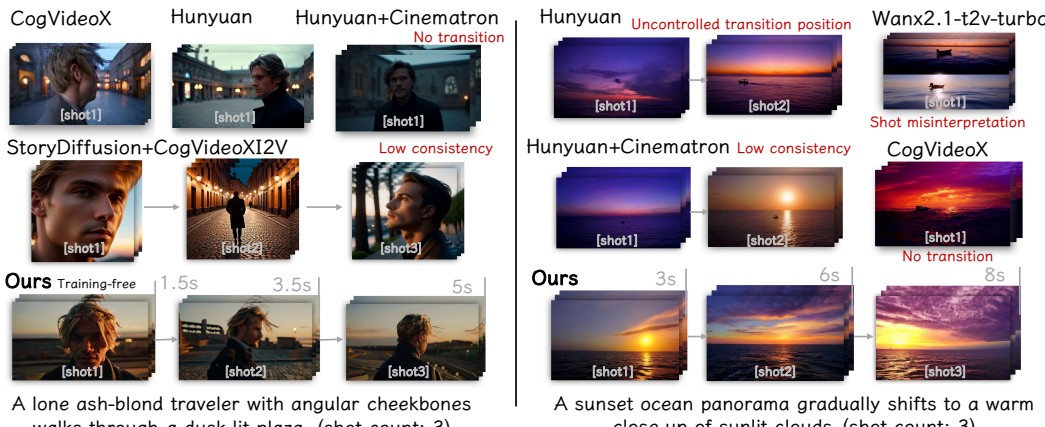

Figure 5: **Qualitative results for different methods.** Our proposed CineTrans outperforms others in transition control while preserving coherence between shots, aligning with film-editing styles. The figure illustrates the shot segmentation results and specified shot count.

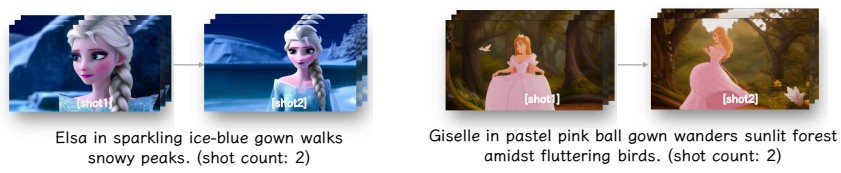

Figure 6: **Results of transferring our method to the customization model.** It enables multi-shot video generation while maintaining consistent character identity.

sequences. Second, unmasked layers enable full-frame token attention, allowing each token to attend to others across different shots, thereby establishing high-level semantic consistency. This facilitates the effective utilization of multi-shot video datasets, thereby generating multi-shot videos that align with film editing. We present the impact of masking different layers in Appendix D.2.

### 4.4 IMPLEMENTATIONS

**Visible-First-Frame Attention.** Beyond the block-diagonal pattern detailed in Section 4.2, we additionally observe that, in certain attention layers, all visual tokens correlate strongly with the first temporal latent. This finding suggests a pronounced reliance on initial video information. To exploit this observation, we introduce a Visible-First-Frame mechanism within the attention layers, augmenting the mask design from Section 4.3 to improve consistency in multi-shot video generation. Further details are provided in Appendix C.1.

**Customization.** Building on our mask mechanism, we guide a diffusion model originally designed for single-shot generation to execute user-specified transitions, thus producing genuine multi-shot videos. For instance, by incorporating LoRA (Hu et al., 2022) weights, we achieve zero-shot generation of multi-shot sequences with enhanced consistency and user-defined styles or character attributes, even though those weights are originally trained on single-shot videos.

## 5 EXPERIMENT

In this section, we present the implementation details, evaluation settings, and results. Section 5.1 describes the details of CineTrans and baselines, which are evaluated on a series of metrics designed for the cinematic multi-shot video generation task. The metrics and results are presented in Section 5.2. Section 5.3 demonstrates that CineTrans performs well due to the components we propose.

Table 1: **Quantitative results**. The best and runner-up are in **bold** and underlined.

| Method | Transition Control Score↑ | Inter-shot Consistency | | | | Intra-shot Consistency | | Aesthetic Quality↑ | Semantic Consistency↑ |
| | | Semantic | | Visual | | Subject↑ | Background↑ | | |
| | | Score↑ | Gap↓ | Score↑ | Gap↓ | | | | |
| StoryDiffusion +CogVideoXI2V | - | 0.5214 | 0.4966 | 0.5660 | 0.3605 | **0.9783** | 0.9713 | 0.6296 | 0.2091 |
| HunyuanVideo +Cinematron | 0.3787 | 0.5631 | 0.3764 | 0.6053 | 0.2855 | 0.9606 | 0.9721 | 0.5978 | 0.2082 |
| HunyuanVideo | 0.2111 | 0.5723 | 0.4075 | 0.5436 | 0.3485 | 0.9476 | 0.9633 | 0.6042 | 0.2064 |
| Wanx2.1-T2V-turbo | 0.2355 | 0.6431 | 0.3002 | 0.6516 | 0.2333 | 0.9332 | 0.9590 | 0.6324 | 0.2046 |
| CogVideoX | 0.0324 | 0.5150 | 0.5915 | 0.6248 | 0.2226 | 0.9310 | 0.9582 | 0.5509 | 0.2061 |
| CineTrans-Unet (Ours) | **0.8598** | **0.8095** | 0.2444 | 0.7247 | **0.1457** | 0.9598 | 0.9725 | 0.5747 | **0.2224** |
| CineTrans-DiT (Ours) | 0.7003 | 0.7858 | **0.1552** | **0.7874** | 0.1901 | 0.9673 | **0.9775** | **0.6508** | 0.2109 |

Table 2: **Quantitative results** for ablation study. The best are in **bold**.

| Method | Transition Control Score↑ | Inter-shot Consistency | | | | Intra-shot Consistency | | Aesthetic Quality↑ | Semantic Consistency↑ |
| | | Semantic | | Visual | | Subject↑ | Background↑ | | |
| | | Score↑ | Gap↓ | Score↑ | Gap↓ | | | | |
| **CineTrans-Unet Ablation** | | | | | | | | | |
| w/o Mask, w/o Tuning | 0 | - | - | - | - | 0.9615 | 0.9702 | **0.5901** | 0.2110 |
| w/o Mask, w/ Tuning | 0.2398 | 0.7900 | 0.3279 | 0.7226 | 0.2148 | 0.9582 | 0.9718 | 0.5711 | 0.2077 |
| w/ Mask, w/o Tuning | 0.6168 | 0.7962 | 0.4336 | **0.8186** | 0.3000 | **0.9616** | 0.9719 | 0.5764 | 0.2196 |
| CineTrans-Unet | **0.8598** | **0.8095** | **0.2444** | 0.7247 | **0.1457** | 0.9598 | **0.9725** | 0.5747 | **0.2224** |
| **CineTrans-DiT Ablation** | | | | | | | | | |
| w/o Mask, w/o Tuning | 0.2051 | 0.5924 | 0.3421 | 0.5274 | 0.3574 | 0.9153 | 0.9523 | 0.6322 | 0.2063 |
| w/o Mask, w/ Tuning | 0.2112 | 0.6532 | 0.3422 | 0.6087 | 0.3312 | 0.9213 | 0.9526 | 0.6345 | 0.2019 |
| w/ Mask, w/o Tuning | 0.6564 | 0.7838 | 0.1772 | 0.7844 | 0.1943 | 0.9618 | 0.9746 | **0.6556** | 0.2093 |
| CineTrans-DiT | **0.7003** | **0.7858** | **0.1552** | **0.7874** | **0.1901** | **0.9673** | **0.9775** | 0.6508 | **0.2109** |

## 5.1 IMPLEMENTATION DETAILS

We implement CineTrans-Unet based on LaVie (Wang et al., 2024b), and the mask mechanism is applied to the last six layers. We finetune the model on Cine250K with a batch size of 128 for 20,000 steps and the learning rate is $1 \times 10^{-4}$. CineTrans-DiT extends Wan2.1-T2V-1.3B (Wan et al., 2025) by integrating the mask mechanism applied to transformer layers 7-28 and is released in two variants. The training-free variant augments the block-diagonal mask with the Visible-First-Frame Attention (Section 4.4). The second variant further applies LoRA fine-tuning (rank=64) with a batch size of 256 for 2,800 steps. All experiments are conducted on NVIDIA A100 GPUs. We also apply LoRA weights to CineTrans-DiT for model customization, as shown in Figure 6.

For baseline comparison, we select three categories: large-scale T2V diffusion model, multi-shot model, and customization model. CogVideoX1.5-5B (Yang et al., 2024), HunyuanVideo (Kong et al., 2024), and Wanx2.1-T2V-turbo[2] leverage large-scale pretraining and thus possess strong semantic understanding. Each of these models is prompted with a general instruction specifying the desired shot count. StoryDiffusion (Zhou et al., 2024) first produces a sequence of semantically consistent images following both the general prompt and individual shot-specific prompts, which CogVideoXI2V (Yang et al., 2024) then expands into a video. As a customization model, Cinematron[3] offers dedicated transition capabilities and employs the same sampling procedure as Hunyuan.

## 5.2 EVALUATION

For comprehensive evaluation, we design 100 hierarchical prompts with transitions using GPT-4o (Achiam et al., 2023), where each initial general description is annotated with shot count and then expanded into shot-level captions, thus constructing a complete prompt. During evaluation, each

---

[2]https://tongyi.aliyun.com/wanxiang/

[3]https://civitai.com/api/download/models/1494601?type=Model&format=SafeTensor

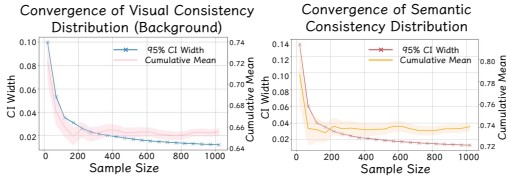 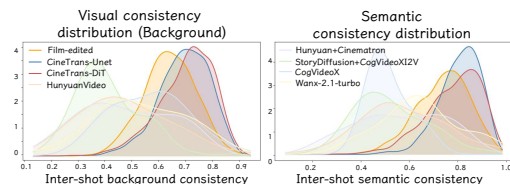

(a) Convergence of consistency distribution.  (b) Consistency distribution for the methods.

Figure 7: **Inter-shot consistency metric details and results.** (a) As the number of samples in the reference set increases, the 95% CI Width and the cumulative mean of subset converge, indicating stability in the reference set's inter-shot consistency distribution. (b) Inter-shot consistency distributions of different methods, CineTrans most closely matches that of the film-edited reference set.

method uses its required prompt format (multi-prompt or single-prompt). To ensure fair comparison, when using only the single general prompt, the shot count is explicitly included in the text.

**Metrics.** We evaluate generated videos along three dimensions: transition control, temporal consistency, and overall quality. For transition control, we assess whether the shot count in the generated video matches the specified count using the Transition Control Score, computed as in Equation 5, where $s_{\text{generated}}$ and $s_{\text{specified}}$ denote the shot counts in the generated video and the prompt, respectively, and $k$ is a hyperparameter controlling the tolerance margin.

$$x = \frac{s_{\text{generated}}}{s_{\text{specified}}} \tag{4}$$

$$\text{Transition Control Score} = \frac{x^k}{e^{k(x-1)}} \tag{5}$$

Second, temporal consistency includes both intra- and inter-shot coherence. For intra-shot, following VBench (Huang et al., 2024), we compute subject and background consistency for each shot and average the results. For inter-shot, we consider both semantics and visuals. Semantically, we compute cosine similarity of ViCLIP (Wang et al., 2023b) features across shots. Visually, following VBench-Long, we compute subject and background similarity between middle frame of shots and average them. Furthermore, as discussed in Section 4.1, overly high inter-shot scores may reflect undesirable pixel-level similarity, which violates the prior of multi-shot video design. To address this limitation, we introduce Consistency Gap as an auxiliary metric, defined as the Jensen-Shannon Distance between the score distribution of generated videos and that of the reference set:

$$\text{JSD}(P, Q) = \sqrt{\frac{1}{2} D\left(P \parallel \frac{P+Q}{2}\right) + \frac{1}{2} D\left(Q \parallel \frac{P+Q}{2}\right)}. \tag{6}$$

Figure 7a illustrates the convergence of the consistency distribution for the reference set, which consists of 1,000 professionally edited multi-shot videos. This metric complements the raw consistency score by quantifying the deviation from natural film-editing style, thus providing a more comprehensive evaluation of video consistency. Finally, following VBench, we assess overall video quality using aesthetic quality and overall consistency, which also evaluate the visual and semantic aspects, respectively.

**Results.** The quantitative comparison is shown in Table 1. Our proposed CineTrans achieves near-perfect cinematic transition control across all prompts. In terms of consistency, CineTrans achieves high scores and closely aligns with film-editing style, which is also shown in Figure 7b. Because aesthetic quality is largely determined by the base model, CineTrans-Unet performs slightly worse in this regard, whereas CineTrans-DiT exhibits superior results.

For qualitative results, as shown in Figure 5, we compare the generated results of different methods. Our method demonstrates a remarkably frame-level transition control capability while preserving coherence across different shots. Even without fine-tuning, CineTrans-DiT exhibits strong performance, demonstrating the transferability of the framework. In contrast, large-scale pretrained models fail to adhere to specified shot counts or fully misinterpret the concept of cinematic transitions. Similarly, both customized models and existing multi-shot models show low temporal consistency.

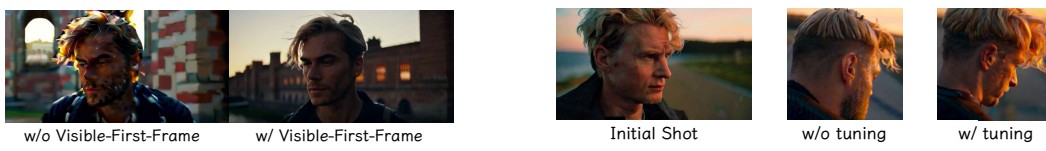

(a) Effects of Visible-First-Frame Attention.     (b) Qualitative comparison of Tuning for 0.4 epoch.

Figure 8: **Qualitative comparison of key components.** (a) Applying the Visible-First-Frame Attention in CineTrans-DiT stabilizes frames. (b) Fine-tuning enhances consistency between shots.

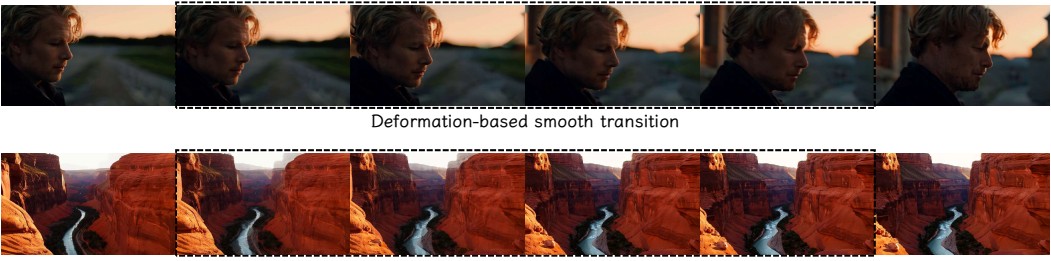

Figure 9: Smoother transitions achieved through soft masking.

### 5.3 ABLATION STUDIES

We conduct ablation studies to assess the impact of the mask mechanism and fine-tuning process. The results in Table 2 demonstrate the effectiveness of our components and the reasonable performance of the training-free variant. In terms of inter-shot visual consistency, the fine-tuned model yields a lower Consistency Score, reflecting increased compositional variation between shots introduced by training on film-edited multi-shot videos. This effect is further corroborated by a reduced Consistency Gap, indicating closer alignment with the film-editing style. The slight decline in Aesthetic Quality after fine-tuning may be attributed to aesthetic domain differences between Cine250K and the original training set of base model. Furthermore, Figure 8 illustrates how fine-tuning and Visible-First-Frame Attention enhance video consistency and enable stable generation.

Additionally, since hard cuts represent the majority of cinematic transitions (99.58%), we implement shot transitions using hard masks. However, there remains a need to explore soft mask strategies to achieve smoother multi-shot video generation, including seamless transitions between frames without visual breaks, as well as fade-in/fade-out effects. We present preliminary results in Figure 9, with further discussion in Appendix D.5. Soft masks improve transition smoothness and consistency but reduce shot transition control, introducing a gap compared to cinematic transitions. Thus, hard masks remain the primary method for shot transitions. Nonetheless, the promising results of the soft mask approach highlight CineTrans's flexibility and robustness, demonstrating its ability to adapt to a wider range of transition styles.

## 6 CONCLUSION

In this paper, we introduce a novel framework CineTrans for cinematic multi-shot video generation and construct a comprehensive dataset Cine250K with detailed shot annotations. Through the analysis of attention maps in video diffusion models, we identify a strong connection between attention probabilities and cinematic transitions. Based on this observation, we propose a novel mask mechanism that enables fine-grained control over cinematic transitions, thus leading to CineTrans framework which transfers successfully in a training-free setting. Extensive experiments validate the effectiveness of CineTrans across multiple evaluation metrics, demonstrating improved transition control, temporal consistency, and overall video quality. Our work demonstrates the potential of diffusion models for multi-shot video generation, offering a new perspective for directly generating movie-like videos and paving the way for future research on controllable video synthesis.

ACKNOWLEDGMENT

The work was supported by Shanghai Artificial Intelligence Laboratory PJ-PRJ24ARC001-20082025. We gratefully acknowledge the support and resources provided, which made this research possible.

ETHICS STATEMENT

This work makes use of video data collected from the publicly available Vimeo platform. We only use videos that are publicly accessible and do not require authentication or bypassing of any access restrictions. The dataset is constructed solely for research purposes, in compliance with Vimeo's Terms of Service, and will not be used for any commercial applications. To mitigate potential privacy concerns, we avoid videos with personally identifiable information and exclude sensitive content. We will not redistribute the raw videos; instead, we provide scripts that allow others to download the data directly from Vimeo if permitted by the platform.

REPRODUCIBILITY STATEMENT

We have made significant efforts to ensure the reproducibility of our work. The source code is provided in the supplementary materials. The dataset construction procedure is described in detail in Section 3 and Appendix A, which enables other researchers to replicate the data preprocessing pipeline. Together, these resources are intended to facilitate the reproduction and verification of our experimental results.

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

# Contents

# A  DETAILS OF DATASET CONSTRUCTION

Section 3 describes the Cine250K construction pipeline. In this section, we provide a detailed account of different stage.

## A.1  METHOD OF SPLITTING AND SHOT SEGEMENTATION

During the Splitting stage, videos are initially segmented using PySceneDetect (Castellano, 2024). In the subsequent gradual-change removal stage, TransNetV2 (Soucek & Lokoc, 2024) is employed to detect and handle gradual transitions, yielding the final shot boundaries and associated shot labels. This pipeline is selected because Pyscenedetect operates exclusively on the CPU, offering greater processing efficiency than Transnetv2; however, for a pre-segmented video, Transnetv2 achieves higher shot segmentation accuracy and can handle gradual transitions more effectively than Pyscenedetect.

For Transnetv2, Figure 10 presents an example of detection. Transnetv2 provides *single-frame-prediction* and *all-frame-prediction*, which are designed to handle hard cuts and gradual transitions, respectively. The *single-frame-prediction* refers to the probability of an individual frame being a transition. In contrast, the *all-frame-prediction* predicts all frames involved in a transition, essentially estimating the probability of a frame being part of a gradual change. We apply threshold-based filtering to exclude frames with a high probability of being transition frames and obtain shot labels. Table 3 presents the threshold settings.

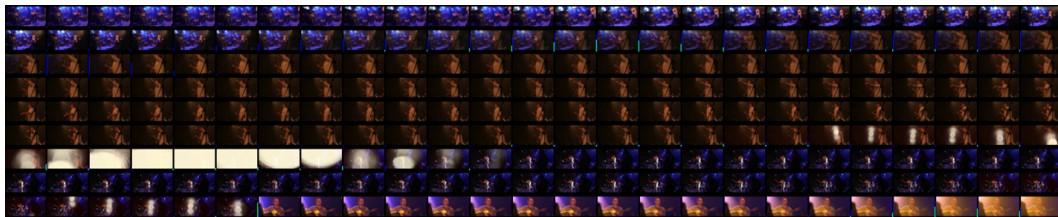

Figure 10: An example of transition detection by Transnetv2. The green line represents the result of *single-frame-prediction*, while the blue line represents the result of *all-frame-prediction*.

Table 3: Threshold settings for dataset construction.

| parameter | value |
|---|---|
| Pyscenedetect splitting | 27 |
| $\alpha$ | 0.9 |
| $\beta$ | 0.7 |
| $\gamma$ | 0.8 |
| Transnetv2  *single-frame-threshold* | 0.45 |
| Transnetv2  *all-frame-threshold* | 0.50 |

Table 4: Shot Segmentation Accuracy of different methods.

| Method | Pyscenedetect | Transnetv2 |
|---|---|---|
| Shot Segmentation Accuracy | 65.50% | 87.00% |

To quantitatively compare accuracy on shot segmentation, we randomly select 200 videos covering multiple categories, each containing multiple shots. We then apply both Pyscenedetect and Transnetv2 for shot segmentation and compare the results manually. A correctly segmented video is assigned 1; otherwise, 0. The shot segmentation accuracy is then calculated accordingly, as demonstrated in Table 4. A specific example is shown in Figure 11.

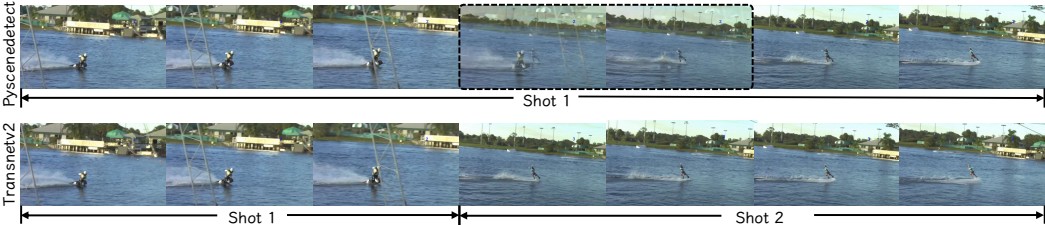

Figure 11: An example of shot segmentation by Pyscenedetect and Transnetv2. In the figure, Transnetv2, after the removing gradual changes step, accurately segments the two shots, while Pyscenedetect fails to identify the gradual changes. The transition frames caused by gradual changes are marked with dashed lines.

## A.2 DETAILS OF STITCHING PHASE

During the Stitching stage, adjacent segments exhibiting high semantic similarity are merged to form a single video, thereby constructing multi-shot video collections. Specifically, we extract semantic features from the first and last frames of each segment using ImageBind (Girdhar et al., 2023) and quantify their similarity via Euclidean distance. For the $i$-th segment $C^i$, we denote the semantic features of its first and last frames as $C^i_{\text{first}}$ and $C^i_{\text{end}}$, respectively, with their distance represented as $\text{dis}(C^i_{\text{first}}, C^i_{\text{end}})$. Based on the distance, video segments are processed sequentially and either stitched or filtered according to the following criteria.

- For a segment $C^i$, if $\text{dis}(C^i_{\text{first}}, C^i_{\text{end}}) > \alpha$, the segment is filtered.
- For $C^i$, if $C^{i-1}$ is absent (either non-existent or filtered), $C^i$ is treated as a video's beginning.
- For $C^i$, if $\text{dis}(C^{i-1}_{\text{end}}, C^i_{\text{first}}) < \beta$ and $\text{dis}(C^{i-1}_{\text{first}}, C^i_{\text{end}}) < \gamma$, then $C^{i-1}$ and $C^i$ are stitched to form a new segment.

Here, $\alpha$ restricts significant changes within a clip, $\beta$ enables stitching of semantically similar transitions or originally continuous segments, and $\gamma$ ensures consistency between the video's beginning and end. After processing all segments sequentially, each resulting segment is considered a complete video, forming a preliminary dataset of videos with shot transitions.

## B STATISTIC OF CINE250K

As we present in Section 3, Cine250K is a carefully curated multi-shot video dataset with detailed captions. This section presents its overall statistics. As shown in Figure 12, the average video duration is 10.75s, and the average caption length is 148.79. Most videos contain 2 to 3 shots. It is important to note that although duration and shot count filtering are applied during data processing, TransnetV2 (Soucek & Lokoc, 2024) is later utilized to remove gradual changes and re-identify shots. As a result, the final shot count and duration do not strictly adhere to the initial filtering criteria. After reidentification, 87.99% of the videos contain 2 to 5 shots. Figure 12 presents the shot distribution for videos with 1 to 10 shots, which represents 99.90% of all videos.

Regarding video categories, following Vimeo's classification, the dataset is divided into 10 categories. Among them, Travel and Documentary have relatively higher proportions. Overall, the distribution of categories is fairly balanced.

In Figure 13, we select several example video-text pairs to demonstrate the specific characteristics of the cinematic multi-shot sequences in the dataset and the style of captions. In Table 5, we compare Cine250K with other datasets. As a multi-shot video dataset with detailed shot labels, the high-quality content of Cine250K can significantly facilitate research and exploration in multi-shot video generation.

Additionally, when Cine250K is used as the training dataset for CineTrans, the distribution of transition point positions and shot lengths is relatively uniform (Figure 14), which enables CineTrans to

Table 5: Comparison of Cine250K and other video-text datasets.

| Dataset | #Videos | Avg video len | Avg text len | Multi-shot | shot label | Aesthetic | Resolution |
|---|---|---|---|---|---|---|---|
| InternVid (Wang et al., 2023b) | 234M | 11.7s | 17.6 | ✗ | - | High | 720P |
| LVD-2M (Xiong et al., 2025) | 2M | 20.2s | 88.7 | ✗ | - | Medium | Diverse |
| OpenVid-1M (Nan et al., 2024) | 1M | N/A | N/A | ✗ | - | High | Diverse |
| Panda70M (Chen et al., 2024b) | 70.8M | 8.5s | 13.2 | ✓ | ✗ | Medium | 720P |
| LLaVA-Video-178K (Zhang et al., 2024) | 178K | $\sim 40.4$s | $\sim 300$ | ✓ | ✗ | High | Diverse |
| Shot2Story20K (Han et al., 2023) | 20K | 16s | 201.8 | ✓ | ✓ | Medium | 720P |
| Shot2Story134K | 134K | N/A | N/A | ✓ | ✓ | Medium | 720P |
| Cine250K (Ours) | 250K | 10.75s | 148.79 | ✓ | ✓ | High | 720P |

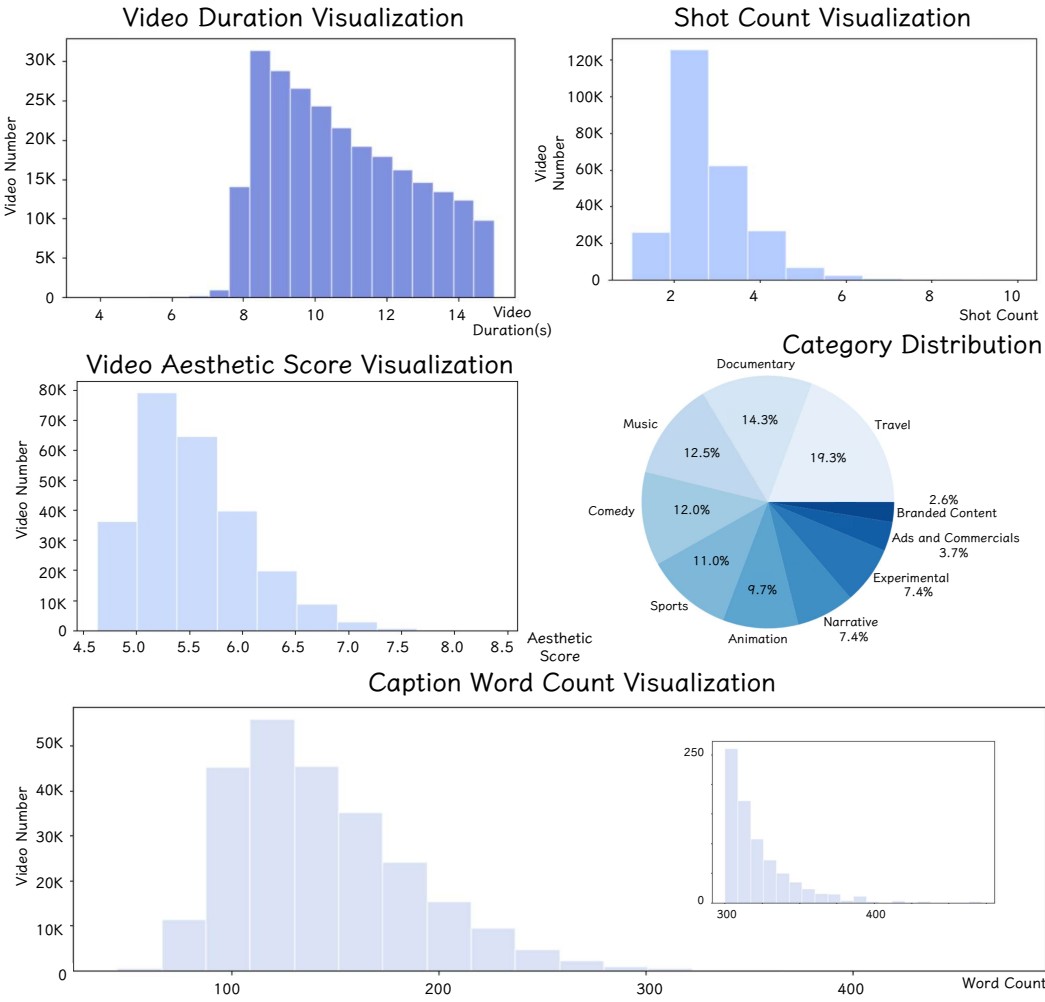

Figure 12: Statistics of Cine250K. To facilitate observation, the figure presents the shot distribution for videos containing 1 to 10 shots. The caption word count distribution only considers data with fewer than 500 words.

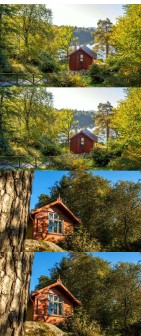

The video showcases a serene and picturesque setting featuring *a small, red wooden cabin* nestled amidst lush greenery. The cabin, with its steeply pitched roof and a chimney, is *surrounded by dense foliage*, including various trees and shrubs, creating a sense of seclusion and tranquility. *A clear lake is visible in the background*, reflecting the surrounding landscape and adding to the peaceful ambiance. The sunlight filters through the leaves, casting dappled shadows on the ground and highlighting the vibrant colors of the foliage. *The camera angle shifts* slightly, offering *different perspectives* of the cabin and its surroundings, emphasizing the natural beauty and calmness of the location.

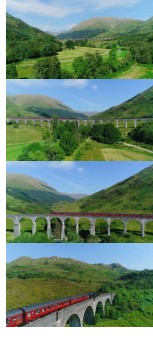

The video *begins with* a wide aerial view of a lush *green valley* surrounded by rolling hills and mountains. A stone viaduct with multiple arches stretches across the valley, and *a red train is seen traveling along the tracks*. The train moves slowly from left to right, crossing the viaduct and moving out of the frame. The background features a clear blue sky with a few scattered clouds, and the landscape is vibrant with greenery. *The scene transitions to a* closer aerial view of *the same red train*, now clearly visible as it crosses the stone viaduct. The train continues its journey from left to right, maintaining a steady pace. The surrounding landscape remains lush and green, with *rolling hills and mountains in the background under a clear blue sky*. The video *concludes with the train still in motion*, gradually moving out of the frame.

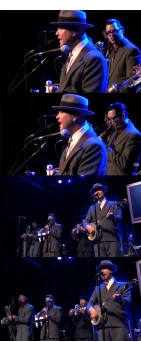

The video features a *live musical* performance on stage, where *a group of musicians* dressed in vintage-style suits and *hats* are playing various instruments. The central figure, wearing a *gray pinstripe suit, white shirt, red tie, and a gray fedora hat*, is holding a microphone and appears to be singing or speaking. Behind him, *another musician is playing a saxophone*, while *a third musician* is playing a trombone. The stage is dimly lit with blue lighting, creating an intimate atmosphere. A banner with the text 'music is life' is visible in the background, emphasizing the theme of the performance. The musicians are deeply engaged in their performance, contributing to the overall ambiance of a live jazz or swing band concert.

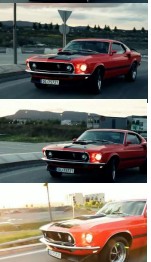

The video features a *red classic muscle car* driving on a road during the late afternoon or early evening. The car, with its headlights on and license plate reading 'DL 75721', is seen from various angles as it moves forward. The *background includes a mix of urban and natural elements*, such as buildings, trees, and mountains under a partly cloudy sky. *The setting sun casts a warm glow over the scene*, enhancing the visual appeal. The car's sleek design and shiny exterior are highlighted as it navigates through the area, showcasing its smooth motion and the picturesque surroundings.

Figure 13: Example video-text pairs in Cine250K.

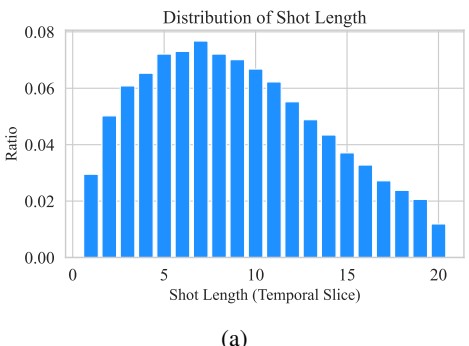

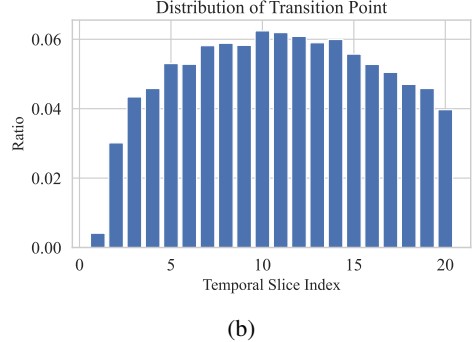

|     |     |
| :-: | :-: |
| (a) | (b) |

Figure 14: Distribution of shot length and transition points in Cine250K as the training dataset. Given the temporal compression of the VAE encoder, temporal slices are used as the unit.

achieve precise frame-level control without timestamp jitter and ensure stability when varying shot lengths are used as conditions.

## C  ADDITIONAL DETAILS OF IMPLEMENTATION

### C.1  VISIBLE-FIRST-FRAME ATTENTION

In Section 4.4, we introduce the implementation detail dubbed Visible-First-Frame Attention, which serves to further stabilize the mask mechanism, particularly within DiT architectures, as demonstrated in Figure 8a. This mechanism is also motivated by our analysis of DiT's attention maps during multi-shot video generation. As illustrated in Figure 15, in certain layers all visual tokens assign high attention probabilities to tokens originating from the first frame (owing to temporal compression in the VAE, this in fact corresponds to the first latent temporal slice), suggesting that the initial frame assumes a special function in the diffusion model's denoising process. In light of

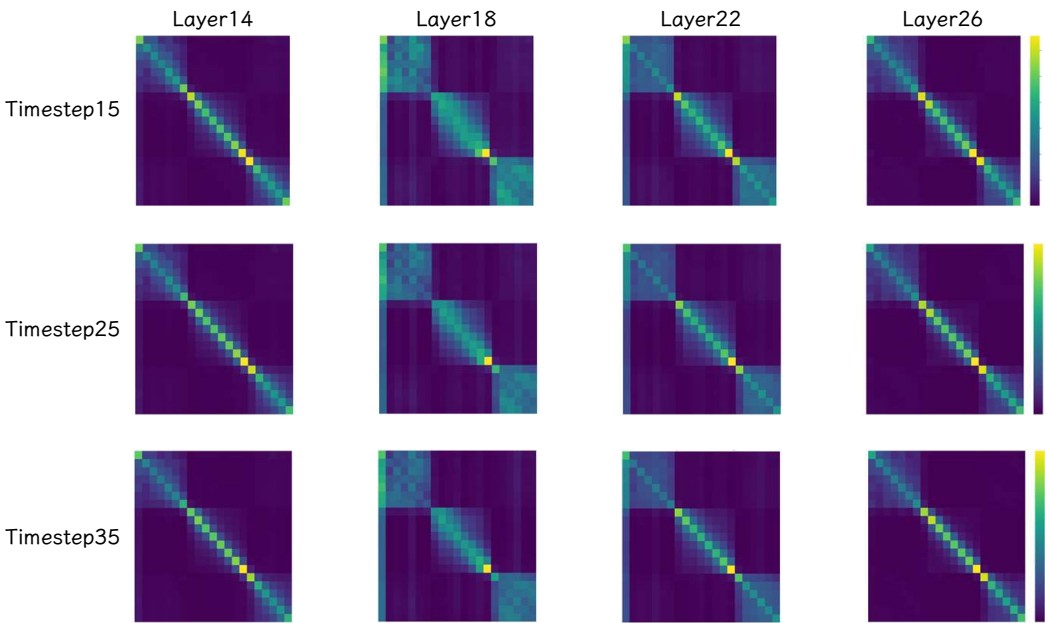

Figure 15: Visualization of the temporal-domain attention maps of Wan2.1 when generating multi-shot videos. Certain layers exhibit a pronounced focus on the first temporal slice, which motivates our Visible-First-Frame Attention mechanism.

this observation, we modify our mask matrix so that its first column is set entirely to zero, thereby effecting the Visible-First-Frame Attention mechanism, which can be formulated as:

$$
\mathcal{M}_{ij} = \begin{cases} 0 & \text{if } j = 1 \text{ or } i, j \in \text{same shot} \\ -\infty & \text{if } j \neq 1 \text{ or } i, j \notin \text{same shot} \end{cases} \tag{7}
$$

## C.2 MULTI-PROMPT

In Section 5.2, we note that some methods employ a multi-prompt strategy at inference, i.e., each shot is prompted by its own text description. CineTrans-DiT also supports this capability. In addition to the primary mask matrix between video tokens, we introduce an additional mask between text and video tokens, following Qi et al. (2025), to enable precise semantic control over each shot. It is worth noting that this mask is applied only to the attention layers governing text–video token interactions, and is distinct from our proposed mask mechanism, which operates on video–video token interactions.

## C.3 TRANSITION POINT SELECTION

In Section 4.3, we introduce the Mask Mechanism that achieves temporal control of shot transitions. During evaluation, shot transition points should be predefined in advance to generate multi-shot videos. To address this, we predefine several suitable shot transition points for different shot counts and randomly select them during inference. However, as mentioned in Appendix B, our training set adequately supports shot transitions at various positions, so random sampling of transition points does not lead to issues such as visual collapse (Figure 16). Nevertheless, for optimal visual performance, including action completeness and overall video quality, uniform shot lengths are preferred, as they avoid extremely short shots.

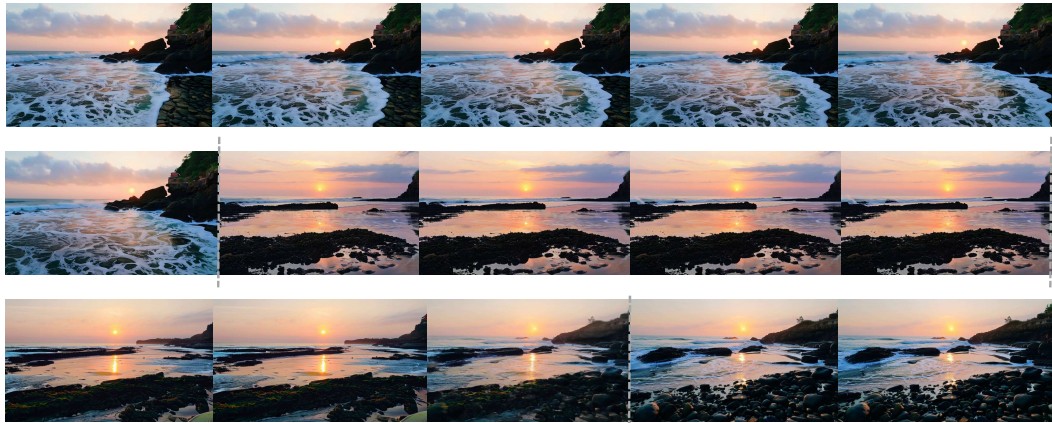

Figure 16: Multi-shot videos with shorter length for last two shots. It can be observed that the visual quality remains intact even when the shot lengths are relatively extreme.

## D  ADDITIONAL RESULTS

### D.1  THE SPECIFIC DETAILS OF THE ATTENTION PROBABILITIES.

In Section 4.2, we explore the frame correlations in the case of cinematic multi-shot video generation in diffusion models, where attention modules in certain layers exhibit a block-diagonal structure, i.e., strong correlations for intra-shot frames and weak correlations for inter-shot frames. In this section, we will discuss the specific details of the attention maps across different architectures, layers, and timesteps.

As for the architecture, we investigate both the temporal-spatial-decoupled framework and the full attention framework. The temporal-spatial-decoupled framework applies the temporal attention module directly to the time sequences, where tokens at different spatial locations do not interact with each other. In contrast, the full attention framework operates on all tokens of the video, allowing correlations across both temporal and spatial dimensions simultaneously. To focus on frame correlations, we group and average the tokens in the full attention framework by the frames before conducting further analysis. In Figure 17 and Figure 18, we use Wang et al. (2024b) and Kong et al. (2024) as representatives of different architectures and present the attention maps across different layers and timesteps in both qualitative and quantitative manners.

As timesteps increase, the discrepancy between intra-shot and inter-shot attention probabilities grows in the temporal-spatial-decoupled framework, whereas it remains stable in the full attention framework. For different layers, the temporal-spatial-decoupled framework exhibits noticeable differences across most layers, whereas the full attention framework shows disparity primarily in the earlier layers. In summary, both frameworks demonstrate significant differences between intra-shot and inter-shot attention probabilities. This finding further substantiates the prevalence of strong intra-shot and weak inter-shot correlations, supporting the proposed approach.

Additionally, we present the inter-shot/intra-shot attention probability ratio for different models and layers, as well the Pearson correlation between this ratio and the distance from transition points (i.e., whether the ratio increases as partition points approach true transitions), with results recorded in Table 6. It can be observed that the inter-shot/intra-shot ratio tends to increase as partition points approach the transition points across different models, further supporting the general presence of the block-diagonal pattern in diffusion models.

Finally, we investigate the potential cause of this pattern by comparing two cases with identical conditions but different seeds: one exhibiting a shot transition and the other only a camera movement (Figure 19). Attention map analysis in Table 7 shows that the case with a shot transition exhibits a stronger block-diagonal pattern, further proving that this attention pattern is strongly correlated with the diffusion model's understanding of shot transitions.

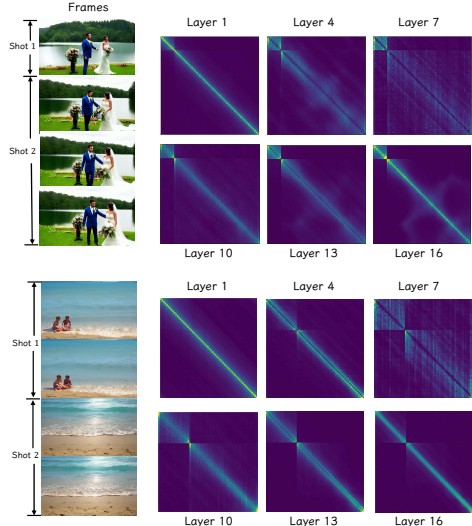

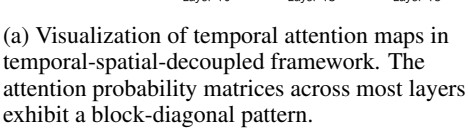

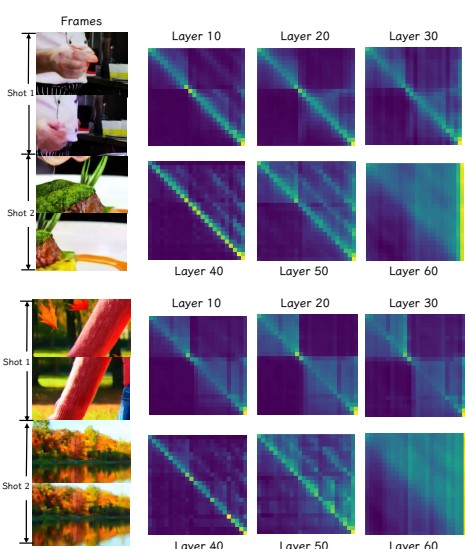

(a) Visualization of temporal attention maps in temporal-spatial-decoupled framework. The attention probability matrices across most layers exhibit a block-diagonal pattern.

(b) Visualization of the averaged attention maps for visual tokens, grouped by frame, in the full attention framework. Earlier layers tend to exhibit a block-diagonal pattern.

Figure 17: The visualization of attention maps between visual tokens from different frames for individual case of multi-shot video generation. Both in the temporal-spatial-decoupled framework and full attention framework, diffusion models exhibit strong intra-shot attention and weak inter-shot attention in certain layers.

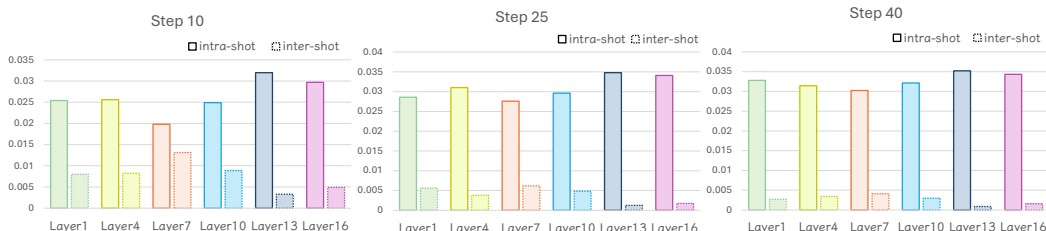

(a) Result of temporal attention probabilities in temporal-spatial-decoupled framework. In most layers, the probabilities of tokens within a shot and those across shots exhibit a noticeable difference, which seems to increase as the denoising process progresses.

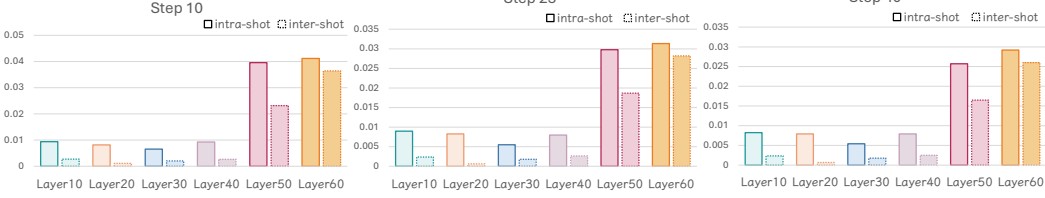

(b) Result of attention probabilities for visual tokens in full attention framework. The difference between the probabilities of tokens within a shot and across shots remains relatively stable during the denoising process, with a tendency to exhibit a larger disparity in the earlier layers.

Figure 18: The average attention probability for intra-shot and inter-shot across diffusion layers and denoising steps. In both framework, the average probability within shots is obvious greater than that between shots for certain layers.

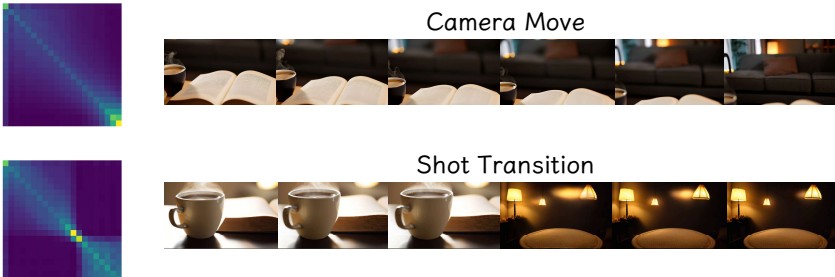

Figure 19: Comparison of attention maps between two cases with identical conditions but different seeds: one exhibiting a shot transition and the other only showing a camera movement.

Table 6: Inter-shot/Intra-shot attention probability ratio and Pearson correlation for different models and layers.

| Model | Intra-shot/Inter-shot attention probability ratio | Pearson correlation |
|---|---|---|
| Wan2.2 (front layer 33%) | 5.6792 | 0.9267,p<0.0001 |
| Wan2.2 (middle layer 33%) | 7.4204 | 0.9233,p<0.0001 |
| Wan2.2 (late layer 33%) | 6.9483 | 0.8496,p<0.0001 |
| Hunyuan (front layer 33%) | 4.8514 | 0.6571,p<0.0001 |
| Hunyuan (middle layer 33%) | 3.4478 | 0.7507,p<0.0001 |
| Hunyuan (late layer 33%) | 1.5194 | 0.4969,p<0.0001 |
| Wan2.1 (front layer 33%) | 8.3925 | 0.7393,p<0.0001 |
| Wan2.1 (middle layer 33%) | 14.1942 | 0.7319,p<0.0001 |
| Wan2.1 (late layer 33%) | 14.6686 | 0.6159,p<0.0001 |

Table 7: Inter-shot/Intra-shot attention probability ratio and Pearson correlation for two cases (with and without transition).

| Model | Intra-shot/Inter-shot attention probability ratio | Pearson correlation |
|---|---|---|
| Case with shot transition | | |
| Wan2.1 (front layer 33%) | 8.3925 | 0.7393,p<0.0001 |
| Wan2.1 (middle layer 33%) | 14.1942 | 0.7319,p<0.0001 |
| Wan2.1 (late layer 33%) | 14.6686 | 0.6159,p<0.0001 |
| Case without shot transition | | |
| Wan2.1 (front layer 33%) | 6.7575 | 0.3462,p<0.001 |
| Wan2.1 (middle layer 33%) | 4.9148 | 0.3378,p<0.001 |
| Wan2.1 (late layer 33%) | 6.7575 | 0.3462,p<0.001 |

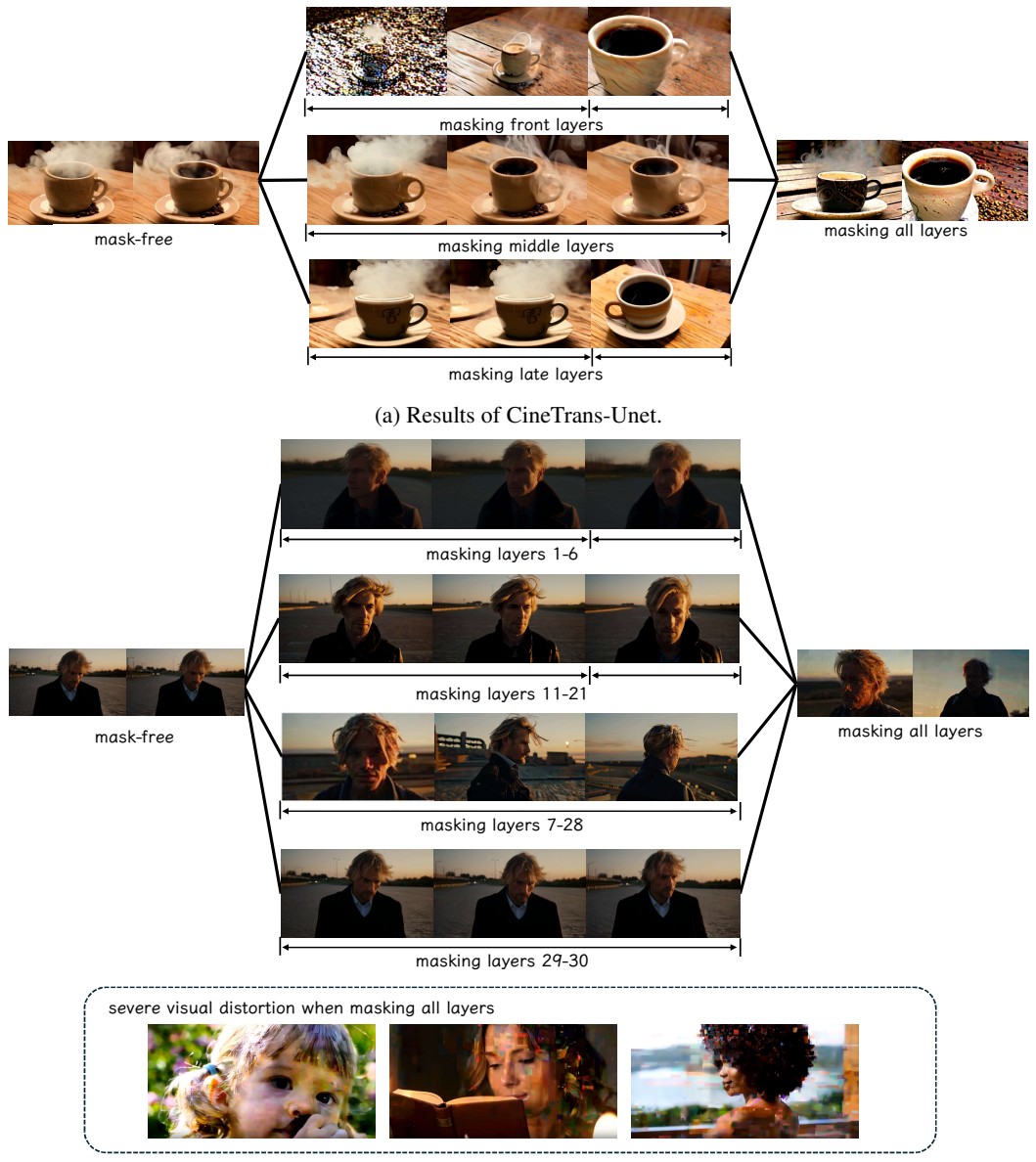

(a) Results of CineTrans-Unet.

(b) Results of CineTrans-DiT.

Figure 20: The results of different mask strategies. Applying the mask to the later layers of the spatial-temporal-decoupled architecture (CineTrans-Unet) and the middle layers of the full attention architecture (CineTrans-DiT) is considered more effective. For the full attention architecture, applying the mask to all layers leads to severe visual distortion, as illustrated in the figure.

## D.2 IMPACT OF MASK LAYERS ON RESULTS

For CineTrans-Unet and CineTrans-DiT, the mask mechanism employs different masking layers, a design choice motivated by observations of the attention maps when diffusion models generate multi-shot videos. In this section, we examine the effects of applying masks to different layers on the quality of the generated videos, thereby demonstrating the rationale behind our masking strategy.

As shown in Figure 20a, CineTrans-Unet exhibits noticeable visual distortions when the mask is applied to all layers or only to the early layers. In severe cases, some parts of the video become indistinct or heavily degraded. When no mask is applied or when the mask is applied only to the

Table 8: Quantitivate results for masking different layers in new different frameworks. The best are in **bold**.

| Method | Transition Control Score↑ | Inter-shot Consistency | | | | Intra-shot Consistency | | Aesthetic Quality↑ | Semantic Consistency↑ |
|---|---|---|---|---|---|---|---|---|---|
| | | Semantic | | Visual | | Subject↑ | Background↑ | | |
| | | Score↑ | Gap↓ | Score↑ | Gap↓ | | | | |
| Ablation for VideoCrafter2 (Unet) | | | | | | | | | |
| Mask front 1/2 layers | 0.0889 | 0.8313 | 0.3630 | 0.8338 | 0.3603 | 0.9594 | **0.9723** | 0.6287 | 0.2205 |
| **Mask late 1/2 layers** | 0.1889 | 0.7739 | **0.1980** | 0.7766 | 0.3416 | **0.9728** | 0.9693 | 0.6258 | **0.2213** |
| Ablation for Wan2.2 (DiT) | | | | | | | | | |
| Mask front 1/2 layers | 0.3811 | **0.9425** | 0.5976 | **0.9380** | 0.5205 | 0.9570 | 0.9624 | 0.6298 | 0.2073 |
| **Mask middle 1/2 layers** | **0.5982** | 0.6752 | 0.2084 | 0.7234 | **0.1458** | 0.9474 | 0.9636 | **0.6395** | 0.2140 |
| Mask late 1/2 layers | 0.0085 | - | - | - | - | 0.9466 | 0.9646 | 0.6305 | 0.1943 |

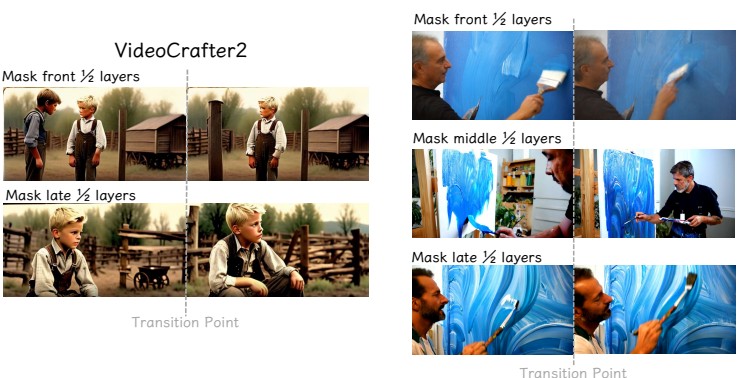

Figure 21: Results of masking different layers in new different frameworks.

middle layers, cinematic transitions do not emerge clearly. Applying the mask to the later layers enables effective control over transitions without significantly affecting visual quality, suggesting it as the optimal strategy. This suggests that, in the spatial-temporal-decoupled architecture, correlations in the earlier layers primarily influence visual quality, while the later layers may regulate the consistency between adjacent frames. Therefore, applying the mask mechanism to the last six layers is demonstrated to be effective.

As shown in Figure 20b, the proportion of layers requiring masking in CineTrans-DiT is larger than in CineTrans-Unet, and both the early and late layers need to retain fully visible attention. Masking all layers to this architecture may lead to low inter-shot consistency and severe visual degradation at the transitions. Conversely, masking only the earlier or later layers fails to effectively guide the transitions. Moreover, when fewer layers are masked, the inter-shot differences are reduced, which deviates from the convention of multi-shot video. These observations suggest that the current strategy of masking the middle layers achieves the optimal balance, enabling precise control over transitions while preserving a reasonable degree of consistency.

To validate the generalizability of our heuristic mask-layer selection, we apply the same approach to other architectures: masking the later half of layers in U-Net and the middle half of layers in DiT. We use VideoCrafter2 Chen et al. (2024a) and Wan2.2 Wan et al. (2025) as base models, with results presented in Table 8. Due to VideoCrafter2's relatively limited duration and representational capacity, the overall Transition Control Scores are relatively low. However, masking later layers still leads to clearer shot-transition behavior. Figure 21 shows that selecting the appropriate masking layer can indeed improve multi-shot video generation performance.

### D.3 QUALITATIVE EVALUATION FOR ABLATION STUDIES

The quantitative results of the ablation studies are presented in Section 5.3, and this section provides a supplementary qualitative analysis.

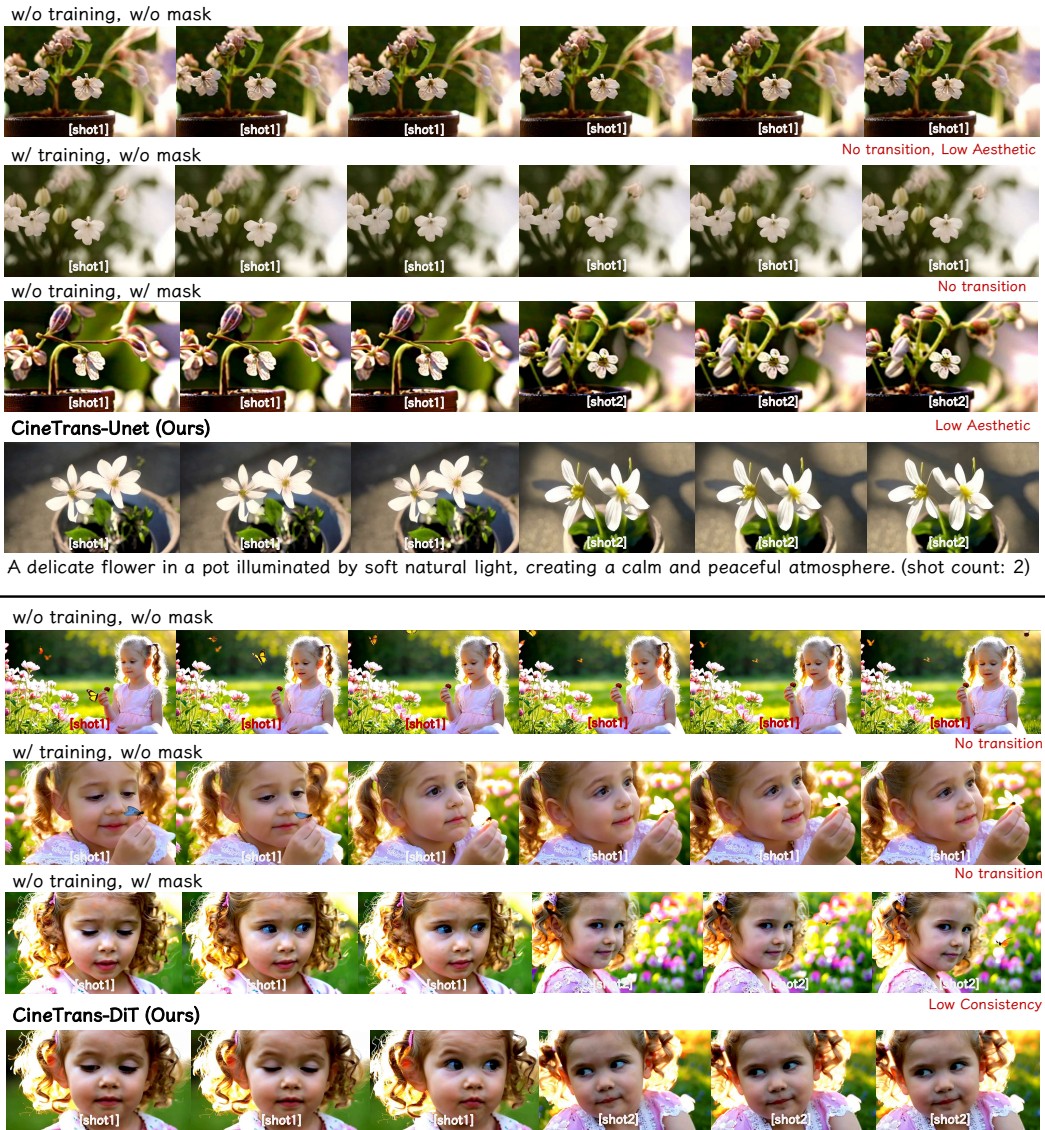

Figure 22: Qualitative results of ablation studies.

As shown in Figure 22, CineTrans can stably control the generation of transitions. In contrast, models without the mask mechanism, whether finetuned or not, are almost incapable of generating cinematic transitions. Direct application of the mask mechanism without further finetuning, i.e., training-free results, does indeed generate transitions. However, further training can help the model produce shot transitions that better align with the film editing style, exhibiting higher aesthetic quality and improved consistency.

## D.4 USER STUDY

To complement the experimental results, we report the findings of our user study in this section. Specifically, we select 20 prompts for user evaluation, where participants rate each result on a scale from 1 to 5. We evaluate different methods from the perspective of Transition Control and Overall Consistency, with the MOS results presented in Table 9. It can be observed that CineTrans also demonstrates strong performance in terms of user preference.

Table 9: Results of User Study

| Model | Transition Control | Consistency |
|---|---|---|
| CineTrans-DiT (ours) | $4.60 \pm 0.50$ | $\mathbf{4.15 \pm 0.75}$ |
| CineTrans-Unet (ours) | $\mathbf{4.75 \pm 0.44}$ | $4.10 \pm 0.72$ |
| StoryDiffusion+CogVideoXI2V | - | $3.95 \pm 0.76$ |
| HunyuanVideo+Cinematron | $3.60 \pm 1.95$ | $3.80 \pm 0.68$ |
| HunyuanVideo | $3.45 \pm 1.88$ | $3.50 \pm 0.76$ |
| CogVideoX | $2.50 \pm 1.24$ | $3.05 \pm 0.60$ |
| Wanx2.1-T2V-turbo | $3.25 \pm 1.77$ | $3.45 \pm 0.69$ |

## D.5 ACHIEVING SMOOTHER TRANSITIONS THROUGH SOFT MASKING

This section investigates soft masking strategies applied to the hard mask mechanism proposed in 4.3, with the aim of assessing whether soft masking can enable smoother shot transitions or higher consistency. Specifically, we explore strategies including time-dependent penalty and timestep-dependent penalty. The following will present the details of these strategies along with the corresponding experimental results.

**Time-Dependent Penalty.** In the hard masking scheme, the mask matrix is initially set to be fully invisible, allowing tokens within each shot to interact for multi-shot video generation. The time-dependent penalty allows token interactions near shot boundaries. Specifically, we initialize the mask matrix using a Gaussian decay based on the distance between frames:

$$\mathcal{M}_{ij} = e^{-\frac{(i-j)^2}{2\sigma^2}}. \tag{8}$$

This is then mapped to the range $[0, -\infty)$, with all tokens within each shot being fully visible subsequently, thus forming a soft masking strategy around shot boundaries. The smoothness of the transition is controlled by the parameter $\sigma$. The evaluation results are presented in Table 10 and Figure 23, where $L$ denotes the sequence length. As the soft mask approaches full visibility, the transition control effect weakens until no shot transition occurs. With the appropriate hyperparameter settings (e.g., $\sigma = L/12$), the visual break at the transition boundary becomes less obvious. However, excessively large $\sigma$ values lead to minimal difference across shot compositions, reducing the content diversity in the multi-shot video and resulting in a more uniform appearance, which explains the decline in semantic consistency.

**Timestep-Dependent Penalty.** In contrast, the Diffusion-Timestep-Dependent Penalty focuses on the timestep of the denoising process, making the invisible region of the mask partially visible during the early denoising steps and gradually approaching full invisibility as the process progresses. Compared to the Time-Dependent Penalty, which focuses on shot boundaries, this soft masking enhances interactions between all inter-shot tokens. As shown in Figure 24, the increased token correlations result in a significant reduction in compositional differences between shots, leaving only a residual boundary transition effect. This comes at the cost of multi-shot content diversity, as reflected by the substantial drop in the Transition Control Score in Table 11. Therefore, the Diffusion-Timestep-Dependent Penalty tends to strengthen inter-shot consistency, while its effect on smoother transitions is less pronounced than that of the time-dependent penalty.

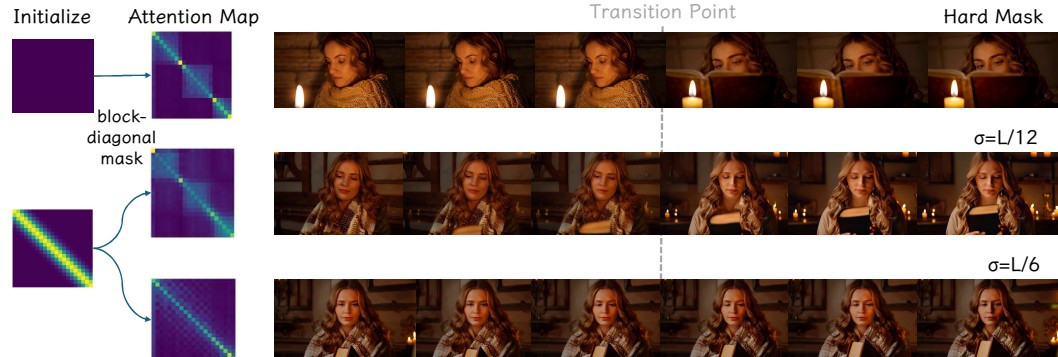

Figure 23: An illustration of the Time-Dependent Penalty.

Table 10: Quantitative results for Time-Dependent Penalty. The best are in **bold**. Smaller $\sigma$ values approach the hard mask.

| Method | Transition Control Score↑ | Inter-shot Consistency | | | | Intra-shot Consistency | | Aesthetic Quality↑ | Semantic Consistency↑ |
|---|---|---|---|---|---|---|---|---|---|
| | | Semantic | | Visual | | Subject↑ | Background↑ | | |
| | | Score↑ | Gap↓ | Score↑ | Gap↓ | | | | |
| *Ablation for CineTrans-DiT (training-free)* | | | | | | | | | |
| $\sigma = L/4$ | 0.0220 | **0.8685** | 0.4261 | 0.8231 | 0.2499 | **0.9629** | **0.9750** | 0.6513 | 0.2025 |
| $\sigma = L/6$ | 0.0700 | 0.8238 | 0.2411 | **0.8248** | 0.2431 | 0.9445 | 0.9606 | 0.6514 | 0.2073 |
| $\sigma = L/12$ | 0.4036 | 0.7987 | 0.2005 | 0.8186 | 0.2139 | 0.9400 | 0.9544 | 0.6562 | 0.2086 |
| Hard Mask | **0.6564** | 0.7838 | **0.1772** | 0.7844 | **0.1943** | 0.9618 | 0.9746 | **0.6556** | **0.2093** |
| *Ablation for CineTrans-DiT (trained)* | | | | | | | | | |
| $\sigma = L/4$ | 0 | - | - | - | - | 0.9541 | 0.9660 | 0.6453 | 0.2004 |
| $\sigma = L/6$ | 0.0455 | **0.7944** | 0.1885 | **0.8377** | 0.3074 | 0.9424 | 0.9598 | 0.6455 | 0.2105 |
| $\sigma = L/12$ | 0.5193 | 0.7902 | 0.1796 | 0.8145 | 0.2575 | 0.9519 | 0.9659 | 0.6489 | 0.2095 |
| Hard Mask | **0.7003** | 0.7858 | **0.1552** | 0.7874 | **0.1901** | **0.9673** | **0.9775** | **0.6508** | **0.2109** |

## D.6 OVERLAPPING AMBIGUOUS SHOT BOUNDARIES

The results in Table 1 demonstrate that our model maintains precise control over shot transitions at specified transition points using the mask mechanism. In this section, we examine the model's performance under overlapping and ambiguous shot boundaries. Specifically, we introduce fluctuations of $\pm t$ frames (temporal slices) around the fixed transition points during inference, which generates overlapping and ambiguous boundaries in the denoising process. Quantitative results are

Table 11: Quantitative results for Timestep-Dependent Penalty. The best are in **bold**. Smaller $t$ values approach the hard mask.

| Method | Transition Control Score↑ | Inter-shot Consistency | | | |
|---|---|---|---|---|---|
| | | Semantic | | Visual | |
| | | Score↑ | Gap↓ | Score↑ | Gap↓ |
| *Ablation for CineTrans-DiT (training-free)* | | | | | |
| $t = 1.0$ | 0 | - | - | - | - |
| $t = 0.5$ | 0 | - | - | - | - |
| Hard Mask | 0.6564 | 0.7838 | 0.1772 | 0.7844 | 0.1943 |
| *Ablation for CineTrans-DiT (trained)* | | | | | |
| $t = 1.0$ | 0 | - | - | - | - |
| $t = 0.5$ | 0.0741 | **0.7864** | 0.2211 | **0.8029** | 0.3655 |
| Hard Mask | **0.7003** | 0.7858 | **0.1552** | 0.7874 | **0.1901** |

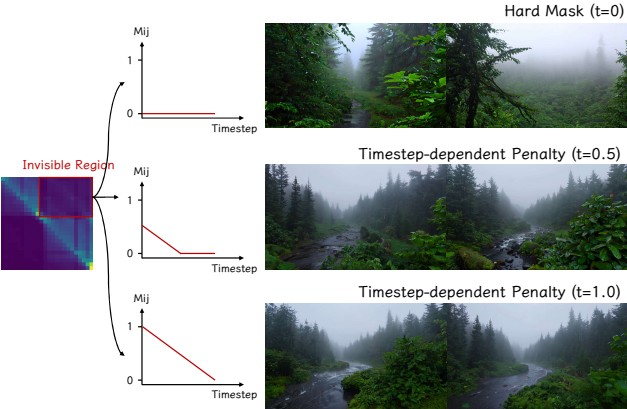

Figure 24: An illustration of the Timestep-Dependent Penalty.

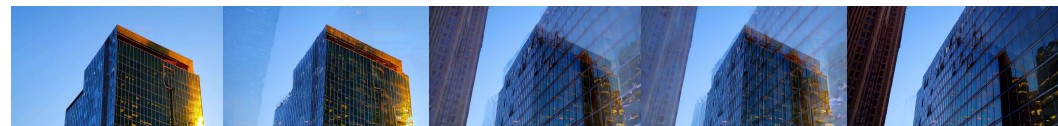

Figure 25: A Result of Overlapping Ambiguous Shot Boundary.

shown in Table 12 and Figure 25. It can be observed that, even under overlapping and ambiguous shot boundaries, the model retains a reasonable Transition Control Score and produces effects resembling fade-in/fade-out transitions to some extent. These results confirm the robustness of the mask mechanism, demonstrating that the model retains effective control even in the presence of ambiguous or overlapping shot boundaries.

# E    EVALUATION

## E.1    PROMPT DETAILS

In Section 5.2, we present a set of prompts generated by GPT-4o (Achiam et al., 2023) to evaluate the performance of multi-shot video generation. Figure 26a provides the prompt used to guide GPT.

Given the long inference time of video generation models, we design 100 prompts for evaluation, covering multiple categories. For these prompts, the transition guidance can be divided into two types. One type explicitly specifies a significant semantic change, such as prompts that include phrases like *The video transition to*, clearly indicating shot changes. The other type implicitly guides the shot transition, where the prompt does not explicitly differentiate content changes between shots, but rather provides an overall description of the video. This type of prompt design accounts for the fact that some cinematic multi-shot videos only involve switching camera angles without significant semantic change. The multi-shot guidance in such cases primarily relies on the prompt's opening phrase: *The multi-shot video consists of {num_shot} shots*. Figure 26b presents examples of these two types of prompts, and Figure 26c shows the word cloud distribution for this set of prompts. Table 13 outlines the categories of the prompts. It is worth noting that, for methods requiring multi-prompt inference, we employ GPT-4o to expand the general prompt into shot-specific captions according to the designated shot count.

## E.2    METRIC DETAILS

In Section 5.2, we establish evaluation metrics from three aspects: transition control, temporal consistency, and overall video quality. This section will provide a detailed explanation of the specific definitions of these metrics.

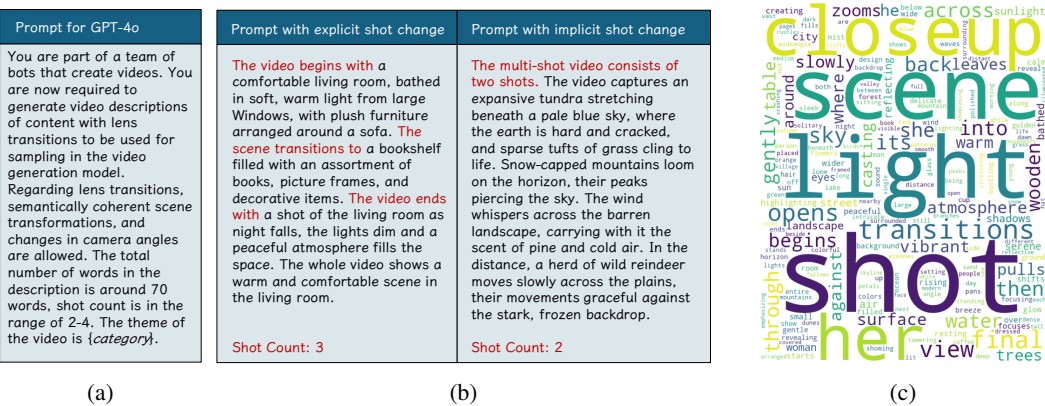

Figure 26: The details of the prompts used for evaluation. (a) Prompt for GPT-4o. The designated video theme vary. (b) Prompt examples for evaluation. (c) Word cloud of prompts for evaluation.

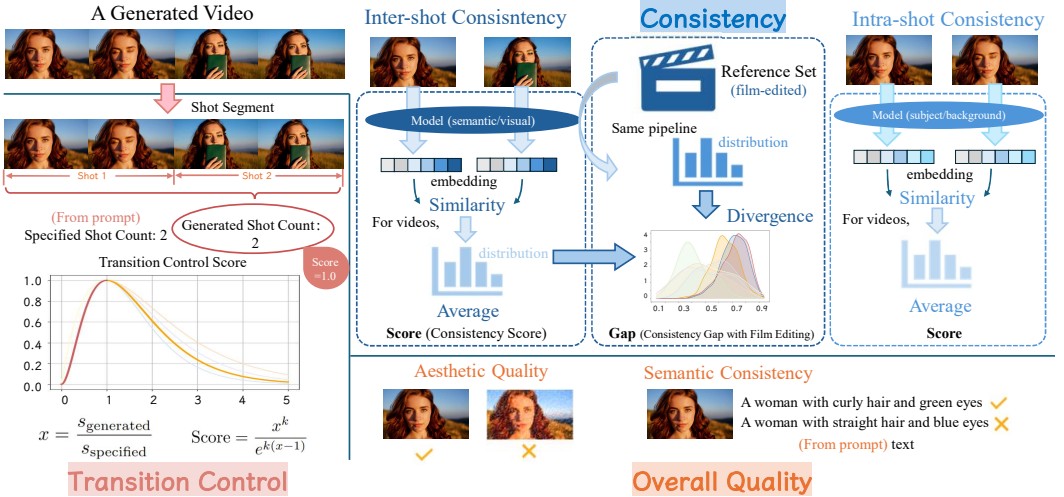

Figure 27: Overview of the metric design. We devise evaluation measures along three complementary dimensions: transition control, temporal consistency, and overall video quality.

Table 12: Quantitative results for Overlapping Ambiguous Shot Boundaries. The best are in **bold**. Smaller $t$ values approach the fixed transition points.

| Method | Transition Control Score↑ | Inter-shot Consistency | | | |
|---|---|---|---|---|---|
| | | Semantic | | Visual | |
| | | Score↑ | Gap↓ | Score↑ | Gap↓ |
| Ablation for CineTrans-DiT (training-free) | | | | | |
| No Mask | 0.2051 | 0.5924 | 0.3421 | 0.5274 | 0.3574 |
| $t = 2$ | 0.2698 | 0.7227 | 0.1918 | **0.8109** | 0.2612 |
| $t = 1$ | 0.4123 | 0.7344 | 0.1846 | 0.8044 | 0.2299 |
| Fixed Transition Points | **0.6564** | **0.7838** | **0.1772** | 0.7844 | **0.1943** |
| Ablation for CineTrans-DiT (trained) | | | | | |
| No Mask | 0.2112 | 0.6532 | 0.3422 | 0.6087 | 0.3312 |
| $t = 2$ | 0.4240 | 0.7526 | 0.1976 | **0.8311** | 0.3082 |
| $t = 1$ | 0.5461 | 0.7302 | 0.1837 | 0.8245 | 0.2764 |
| Fixed Transition Points | **0.7003** | **0.7858** | **0.1552** | 0.7874 | **0.1901** |

Table 13: Details of prompt categories for evaluation.

| category | count |
|---|---|
| scenary | 34 |
| architecture | 10 |
| human | 25 |
| object | 31 |

**Transition Control.** For transition control, we define the Transition Control Score to measure whether the shot count in the generated video aligns with the specified count, as formulated in Equation 5.

Figure 27 visualizes the calculation method for the Transition Control subpanel. In practice, when $x < 1$, $k$ is set to 2, and when $x \geq 1$, $k$ is set to 1.6. Given that the prompts used for evaluation all specify multiple transitions, the score is set to 0 when the generated video consists of a single shot. When the shot count in the generated video matches the specified value, the score is recorded as 1. More generally, the score is determined based on the absolute difference from the specified value.

**Temporal consistency.** In terms of temporal consistency, we consider both intra-shot consistency and inter-shot consistency. Intra-shot consistency treats each shot as a separate video and calculates the metric between adjacent frames using a method similar to that in VBench (Huang et al., 2024). The focus of this section is on inter-shot consistency. As for frame extraction, we use the middle frame of each shot for calculation. However, if the video does not generate multiple shots, inter-shot consistency cannot be evaluated. As shown in Table 1, the original LaVie (Wang et al., 2024b) lacks the ability to generate transitions, and therefore its inter-shot consistency metric does not have a corresponding value.

Regarding the metric definition, inter-shot consistency cannot directly serve as the final evaluation metric. In multi-shot video generation, the goal is to ensure that the generated video aligns with the editing style of film-edited videos. High consistency would imply pixel-level similarity, which may contradict the multi-shot nature of real video editing. To address this, we extract 1000 film-edited videos as a validation dataset and compute their inter-shot consistency as a reference set. The final metric is then determined by the Jensen-Shannon Distance (JSD) between the inter-shot consistency of the generated video and that of film-edited videos, as shown in Equation 6. A lower JSD indicates a closer alignment with the reference distribution.

Videos whose inter-shot consistency distribution aligns with film editing styles are considered to exhibit higher inter-shot consistency performance, while those deviating from film editing practices are regarded as having lower performance. This defines the evaluation metric based on inter-shot consistency.

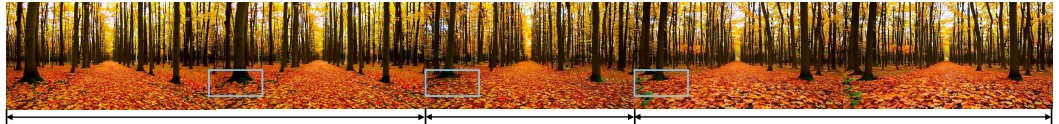

Figure 28: Failure case with similar composition consistency, which probably results from insufficient training.

## F    LIMITATION

### F.1    FAILURE CASE

The role of the mask mechanism in controlling the occurrence of transitions remains relatively stable. Nevertheless, in occasional cases (particularly under the training-free setting), the compositions of different shots may become overly similar, as illustrated in Figure 28. Although such content changes are perceptible to the human eye, the shot segmentation model does not recognize them as multi-shot videos due to the high compositional similarity, and the resulting visual experience also deviates from the film editing style. These cases can be attributed to insufficient training or the absence of such data types in the training set, which prevents the model from generating videos of the same scene with varied compositions. Potential improvements may come from expanding the dataset or further training.

### F.2    FUTURE WORK

Although CineTrans has achieved promising results in cinematic multi-shot video generation using the mask mechanism, there are still limitations to be addressed, along with several promising directions for future research.

- While the mask mechanism enables control over the occurrence of shots, the specification of camera viewpoints could be made more controllable, for example, enabling changes in shooting perspective within a scene that are more consistent with the film production pipeline. Therefore, achieving higher content consistency and more precise control over camera viewpoints will be a key future direction, potentially requiring the incorporation of 3D information as prior knowledge.
- Multi-shot videos could also be extended to greater lengths. Incorporating auto-regressive generation into the video generation pipeline represents a promising approach for producing longer multi-shot videos.
- Fine-grained consistency presents an area for further improvement. While our method demonstrates strong overall consistency, achieving fine-grained alignment across shots remains challenging due to the intricate spatial understanding required. This limitation is primarily due to the lack of detailed background and scene context annotations in the training data. Future work will focus on incorporating more granular annotations and leveraging advanced model designs to address this challenge.

## G    USE OF LARGE LANGUAGE MODELS

We clarify the involvement of large language models (LLMs) in the preparation of this work. LLMs are not used for research design, methodological decisions, or experimental analysis. Their use is limited to two aspects: (i) improving the clarity and readability of the manuscript through language editing, and (ii) generating evaluation prompts during the assessment phase, as explicitly noted in Section 5.2. All substantive research contributions, including the conception of the problem, model design, implementation, and analysis, are conducted entirely by the authors.

