# OpenReview forum: "CineTrans: Learning to Generate Videos with Cinematic Transitions via Masked Diffusion Models"
_ICLR.cc/2026/Conference — ICLR 2026 Poster_

### Official Review · Reviewer_wQCP · 2025-10-29

**Soundness:** 2
**Presentation:** 2
**Contribution:** 3
**Rating:** 4
**Confidence:** 3

**Summary:**

The paper introduces CineTrans, a diffusion-based framework for generating coherent multi-shot videos with cinematic transitions, aiming to bridge the gap between single-shot and multi-shot video generation prevalent in recent works. To support this, the authors construct Cine250K, a large-scale, multi-shot video-text dataset with detailed shot annotations and hierarchical textual descriptions. CineTrans leverages an observed block-diagonal attention pattern for shot boundaries in diffusion models, and implements a mask-based control mechanism that enables frame-level control over cinematic transitions—even in a training-free regime. The approach is empirically validated against several strong baselines, showing superior transition control, temporal consistency, and overall quality via both standard and custom metrics.

**Strengths:**

The work introduces a novel mechanism that discovers and leverages a block-diagonal pattern in attention maps for transition control, directly inspiring an improved architecture; it further uses a mask-based strategy to align model internals with multi-shot video structure, enabling precise, shot-wise editing. A new large-scale dataset, Cine250K, fills a key gap with rich annotations (including frame-level labels and semantic stitching) and follows film-editing conventions. Extensive experiments provide thorough ablations and comparisons on transition control, intra- and inter-shot consistency, and aesthetic quality, complemented by insightful visual analyses. The approach transfers cleanly to customized or pre-trained diffusion backbones and improves both transition control and consistency in training-free and fine-tuned variants, with clear, intuitive examples and visual ablations highlighting advantages over existing methods.

**Weaknesses:**

1. Lack of Analysis on Limiting Scenarios: The paper demonstrates strong results on curated prompts and the Cine250K distribution, but does not critically examine or quantify limitations outside this scope, e.g., severe domain shifts, failure cases, or fundamental breakdowns of mask-based control when transition points are ambiguous or overlap.
2. Limited Theoretical Rigor or Insights: While the empirical demonstration of attention map patterns is clear (see Figure 4), the work lacks theoretical analysis or ablation to clarify why these attention correlations emerge (e.g., role of architecture, dataset, or objective). Are these properties universal, or dataset/model-dependent? This affects reproducibility and generalizability.

**Questions:**

1. Robustness to Ambiguous or Overlapping Shot Boundaries: How does CineTrans perform when shot boundaries are ambiguous or overlap semantically/visually (e.g., with crossfade-like transitions)? Is the mask mechanism brittle or robust under these settings?
2. Mask Relaxation Variants: Have the authors experimented with soft masks or probabilistic boundary definitions? Are there gains in aesthetic quality, or does this impair transition control?
3. Aesthetic Quality Decline After Fine-Tuning: What is the hypothesized cause of the decline in Aesthetic Quality (Table 2) after fine-tuning? Is it dataset-related or architectural?

---

> ### Author Response · Authors · 2025-11-19
> **Response to Reviewer wQCP (1/4)**
>
> *We sincerely appreciate your comprehensive feedback and assessment of our work. We have carefully addressed each point raised, supplemented with additional analyses and clarifications, and revised the manuscript accordingly to address all concerns.*
>
> #  &nbsp;
>
> > W1 (Partial): Lack of Analysis on Limiting Scenarios: The paper demonstrates strong results on curated prompts and the Cine250K distribution, but does not critically examine or quantify limitations outside this scope, e.g., severe domain shifts, failure cases, .......
>
> We appreciate the reviewer’s critical attention to the analysis of out-of-scope scenarios. We address these points with supplementary analysis and experiments as follows:
> - **Domain Shift Analysis: We conduct additional experiments on 5-shot video generation** (not included in the training set) to evaluate domain generalization, with results presented in Table A. As our task uses open-domain prompts, 5-shot videos serve as a representative domain shift scenario. While metrics show a modest decline compared to 2-4-shot generations, performance remains at a high level, demonstrating the model’s reasonable generalization ability across different shot count domains.
> - **Failure Case Analysis**: Our main failure cases involve excessive compositional similarity between consecutive shots (predominantly in the training-free setting), leading to indistinct transitions that resemble "transition jitter." This phenomenon is also discussed in Appendix G.1 of our paper. Such cases are rare in fine-tuned models, and addressing them would require finer-grained annotations or designs that better balance shot consistency and compositional diversity, which is an important direction for future work.
>
> ###  &nbsp;
>
> **Table A. Quantitative results for 5-shot video generation.**
>
> | Method                   | domain  | Transition Control Score(↑) | Inter-shot Consistency |                 |                 |               | Intra-shot Consistency |               | Aesthetic(↑) | Semantic Consistency(↑) |
> | ------------------------ | ------- | --------------------------- | ---------------------- | --------------- | --------------- | ------------- | ---------------------- | ------------- | ------------ | ----------------------- |
> |                          |         |                             | Semantic Score(↑)      | Semantic Gap(↓) | Visual Score(↑) | Visual Gap(↓) | Subject(↑)             | Background(↑) |              |                         |
> | CineTrans-DiT (trained)  | ORI     | 0.7003                      | 0.7858                 | **0.1552**      | **0.7874**      | 0.1901        | **0.9673**             | **0.9775**    | 0.6508       | 0.2109                  |
> |                          | 5-shots | 0.5866                      | 0.8237                 | 0.2855          | 0.7728          | 0.1906        | 0.9459                 | 0.9675        | **0.6677**   | 0.2113                  |
> | CineTrans-Unet (trained) | ORI     | **0.8598**                  | 0.8095                 | 0.2444          | 0.7247          | **0.1457**    | 0.9598                 | 0.9725        | 0.5747       | **0.2224**              |
> |                          | 5-shots | 0.7577                      | **0.8469**             | 0.3816          | 0.7291          | 0.2381        | 0.9613                 | 0.9770        | 0.595        | 0.2194                  |

---

> ### Author Response · Authors · 2025-11-19
> **Response to Reviewer wQCP (2/4)**
>
> > W1 & Q1: Lack of Analysis on Limiting Scenarios: The paper demonstrates strong results on curated prompts and the Cine250K distribution, but does not critically examine or quantify limitations outside this scope, e.g., ...... or fundamental breakdowns of mask-based control when transition points are ambiguous or overlap.
> >
>  >Robustness to Ambiguous or Overlapping Shot Boundaries: How does CineTrans perform when shot boundaries are ambiguous or overlap semantically/visually (e.g., with crossfade-like transitions)? Is the mask mechanism brittle or robust under these settings?
>
> We appreciate the reviewer’s insightful question on robustness to ambiguous or overlapping boundaries and present additional analysis and experiments in response.
> - **Robustness to Ambiguous/Overlapping Shot Boundaries: Our model maintains sufficient robustness under these settings**, even enabling effects approximating crossfade-like transitions. Specifically, we introduce fluctuations of ±$t$ frames (temporal slices) to the originally fixed transition points during inference, creating overlapping and ambiguous boundaries in the denoising process. Quantitative results are shown in Table B (with additional details and visualizations in the revised Appendix E.6). While the Transition Control Score decreases slightly due to the crossfade-like transition design, it remains well above the "no mask" baseline, confirming that the mask mechanism is not brittle and the model retains effective control under ambiguous/overlapping shot boundaries.
>
> ###  &nbsp;
>
> **Table B. Quantitative results for ambiguous/overlapping shot boundaries (complete results are provided in the revised Appendix E.6).**
>
> | Method                   | setting | Transition Control Score(↑) | Inter-shot Consistency |                 |                 |               |
> | ------------------------ | ------- | --------------------------- | ---------------------- | --------------- | --------------- | ------------- |
> |                          | $t$     |                             | Semantic Score(↑)      | Semantic Gap(↓) | Visual Score(↑) | Visual Gap(↓) |
> | CineTrans-DiT (tranined) | ORI     | **0.7003**                  | **0.7858**             | **0.1552**      | 0.7874          | **0.1901**    |
> |                          | 1       | 0.5461                      | 0.7302                 | 0.1837          | 0.8245          | 0.2764        |
> |                          | 2       | 0.4240                      | 0.7526                 | 0.1976          | **0.8311**      | 0.3082        |
> |                          | no mask | 0.2112                      | 0.6532                 | 0.3422          | 0.6087          | 0.3312        |

---

> ### Author Response · Authors · 2025-11-19
> **Response to Reviewer wQCP (3/4)**
>
> > W2: Limited Theoretical Rigor or Insights: While the empirical demonstration of attention map patterns is clear (see Figure 4), the work lacks theoretical analysis or ablation to clarify why these attention correlations emerge (e.g., role of architecture, dataset, or objective). Are these properties universal, or dataset/model-dependent? This affects reproducibility and generalizability.
>
> Thanks for highlighting the importance of theoretical rigor. We provide additional analyses to clarify the origins of the block-diagonal attention pattern from architecture, dataset, and shot transition perspectives, and we have revised the manuscript accordingly (Appendix E.1).
> - **From a theoretical perspective**, the block-diagonal pattern reflects strong correlations within shot and weak correlations across shots, since attention probabilities inherently measure token-to-token correlation. This behavior aligns with the structural property of multi-shot videos, where intra-shot frames exhibit pixel-level correspondence while inter-shot frames share only high-level semantic relations. We also provide a conceptual analysis in Section 4.2 of the paper.
> - **Architecture & Dataset Independence**. We have already provided visual (Figure 14) and quantitative (Figure 15) analyses in Appendix E.1, comparing the temporal-spatial-decoupled framework with the full attention framework and demonstrating that the block-diagonal pattern persists in multi-shot scenarios across different architectures. To further validate, we compute two metrics for multi-shot cases in different models trained on different datasets: (1) intra-shot/inter-shot attention correlation ratios across layers; (2) Pearson correlation between this ratio and distance from transition points (i.e., whether the ratio rises as partition points approach true transitions). Table C confirms all models exhibit the block-diagonal pattern (large r-values), proving **it is not dataset/model-dependent.**
> - **Shot Transitions**. Additionally, we compare two identical-condition cases from Wan2.1: one with shot transitions, the other with only camera movement (Table C). The non-transition case shows a smaller Pearson coefficient and thus weaker block-diagonal pattern, which confirms **the pattern is strongly tied to shot transitions.**
>
> ###  &nbsp;
>
> **Table C. Intra-shot/inter-shot attention correlation ratios and Pearson coefficients for different cases in multi-shot scenarios.**
>
> | Model                                        | intra-shot/inter-shot correlation ratio | Pearson coefficients |
> | -------------------------------------------- | --------------------------------------- | -------------------- |
> | Wan2.2 (front layer 33%)                     | 5.6792                                  | 0.9267,p<0.0001      |
> | Wan2.2 (middle layer 33%)                    | 7.4204                                  | 0.9233,p<0.0001      |
> | Wan2.2 (late layer 33%)                      | 6.9483                                  | 0.8496,p<0.0001      |
> | Hunyuan (front layer 33%)                    | 4.8514                                  | 0.6571,p<0.0001      |
> | Hunyuan (middle layer 33%)                   | 3.4478                                  | 0.7507,p<0.0001      |
> | Hunyuan (late layer 33%)                     | 1.5194                                  | 0.4969,p<0.0001      |
> | Wan2.1 (front layer 33%)                     | 8.3925                                  | 0.7393,p<0.0001      |
> | Wan2.1 (middle layer 33%)                    | 14.1942                                 | 0.7319,p<0.0001      |
> | Wan2.1 (late layer 33%)                      | 14.6686                                 | 0.6159,p<0.0001      |
> | Wan2.1_no_transition_case (front layer 33%)  | 6.7575                                  | 0.3462,p<0.001       |
> | Wan2.1_no_transition_case (middle layer 33%) | 4.9148                                  | 0.3378,p<0.001       |
> | Wan2.1_no_transition_case (late layer 33%)   | 6.7574                                  | 0.3222,p<0.001       |

---

> ### Author Response · Authors · 2025-11-19
> **Response to Reviewer wQCP (4/4)**
>
> > Q2: Mask Relaxation Variants: Have the authors experimented with soft masks or probabilistic boundary definitions? Are there gains in aesthetic quality, or does this impair transition control?
>
> Thank you for your interest in mask relaxation variants. We have explored soft masking (with probabilistic boundary definitions) and present our findings below.
> - As suggested by Reviewer 6pc7, we implement soft masking via a time-dependent penalty, initializing masks with a Gaussian-decayed banded matrix to enable token interactions near shot boundaries and thus achieving probabilistic boundary effects. Quantitative results are provided in Table D (with full details in the revised Appendix E.5).
> - As soft masks approach full visibility, transition control weakens slightly, accompanied by a minor drop in aesthetic quality. This is likely because softer transitions blur shot boundaries and the overlapping transition frames between shots impact overall aesthetic perception.
>
> ###  &nbsp;
>
> **Table D. Quantitative results for soft masking (complete results are provided in the revised Appendix E.5. $\sigma$ denotes the bandwidth, with smaller values closer to hard mask).**
>
> | Method                     | setting       | Transition Control Score(↑) | Aesthetic Score(↑) |
> | -------------------------- | ------------- | --------------------------- | ------------------ |
> | CineTrans-DiT (trained)    | ORI           | **0.7003**                  | **0.6508**         |
> | **Time-dependent Penalty** | $\sigma$=L/12 | 0.5193                      | 0.6489             |
> |                            | $\sigma$=L/6  | 0.0455                      | 0.6455             |
> |                            | $\sigma$=L/4  | 0                           | 0.6453             |
>
> ##  &nbsp;
>
> > Q3: Aesthetic Quality Decline After Fine-Tuning: What is the hypothesized cause of the decline in Aesthetic Quality (Table 2) after fine-tuning? Is it dataset-related or architectural?
>
> Thank you for your question about the aesthetic quality decline post-fine-tuning. As briefly noted in Section 5.3 of the main text, we initially attribute this decline to the distribution gap between the Cine250K dataset and the base model’s original training data. To further validate this, we collect a small-scale movie dataset (10K) with more rigorous aesthetic filtering, and sample outputs from early-stage fine-tuning to track aesthetic quality changes. The results are presented in Table E. The slight upward trend in aesthetic scores during early training further confirms that the aesthetic quality decline observed in the main results is dataset-related.
>
> ###  &nbsp;
>
> **Table E. Impact of different training datasets on Aesthetic Quality.**
>
> | Model                                   | Aesthetic Quality |
> | --------------------------------------- | ----------------- |
> | CineTrans-DiT (training-free)           | 0.6551            |
> | CineTrans-DiT (trained)                 | 0.6502            |
> | CineTrans-DiT (Movie-trained-2000steps) | 0.6554            |
> | CineTrans-DiT (Movie-trained-4000steps) | 0.6567            |
> | CineTrans-DiT (Movie-trained-6000steps) | **0.6572**        |
>
> #  &nbsp;
>
> *Thank you for your thoughtful feedback and constructive comments. We have carefully addressed your concerns. We believe the revisions have strengthened our paper and better highlight its technical contributions. If you have any further questions, we would be happy to address them during the rebuttal window. Thank you again for your valuable input!*
>
> #  &nbsp;
>
> Best regards,
> Authors

---

> ### Author Response · Authors · 2025-11-26
> **Follow-up on Rebuttal**
>
> Dear Reviewer,
>
> Thank you once again for your valuable suggestions that have helped us improve the manuscript.   As the rebuttal period is nearing its end, we would like to kindly check whether our responses have sufficiently addressed your questions and concerns.   If not, we would be glad to provide further clarifications or revisions.   If they have, we would greatly appreciate it if you could consider raising the score accordingly.
>
> Thank you again for your constructive and insightful feedback.
>
> Best regards,
> The Authors

---

> > ### Comment · Reviewer_wQCP · 2025-11-27
> > **response**
> >
> > Thank you for your detailed rebuttal, which has effectively addressed my concerns. I will raise my score accordingly.

---

> > > ### Author Response · Authors · 2025-11-27
> > > **Appreciation for the Updated Score**
> > >
> > > Dear Reviewer,
> > >
> > > Thank you very much for carefully considering our rebuttal and for raising your score.
> > >
> > > We sincerely appreciate your thoughtful comments and the time you devoted to evaluating our work.
> > >
> > > Your updated assessment truly means a lot to us.
> > >
> > > Best regards, Authors

---

### Official Review · Reviewer_ZvZp · 2025-11-01

**Soundness:** 3
**Presentation:** 4
**Contribution:** 3
**Rating:** 6
**Confidence:** 3

**Summary:**

This paper proposes a novel framework for generating coherent multi-shot videos with film-style transitions by introducing a block-diagonal attention mask and a “Visible-First-Frame” mechanism in video diffusion models, enabling precise shot boundaries and stable temporal consistency. Additionally, this paper constructs a 250K video–text dataset with frame-level shot labels to support video diffusion models in generating cinematic transitions and maintaining inter-shot consistency. This paper proposes a series of comprehensive metrics to evaluate the results of multi-shot video generation.

**Strengths:**

1. This paper contributes a large multi-shot dataset of 250K videos with frame-level shot boundaries and hierarchical captions.
2. The proposed method is simple and easy to follow.
3. The paper is well-structured and easy to read.

**Weaknesses:**

1. The method in this paper only adds a mask mechanism between shots to ensure content consistency, but this makes it difficult to maintain fine-grained consistency across different shots, especially for background regions, as shown on the left side of Figure 5. Table 1 (Intra-shot Consistency) also demonstrates that the improvement in consistency achieved by this method is limited compared to the baseline.
2. Although the paper claims to achieve cinematic transitions, the proposed method only supports hard cuts, which limits its range of applications.

**Questions:**

1. Using the mask strategy requires knowing between which frames a shot transition occurs. How does this paper determine the frame length of different shots when generating multi-shot videos? Is it manually set, or controlled by hyperparameters? If it is manually set, how do you ensure that a complete action can be generated within a single shot without being interrupted?

---

> ### Author Response · Authors · 2025-11-19
> **Response to Reviewer ZvZp (1/3)**
>
> *Thank you for the detailed review and for recognizing the strengths of our work. We appreciate your constructive comments and thoughtful questions, which help us further clarify the design choices and contributions of our approach.*
>
> #  &nbsp;
>
> > W1: The method in this paper only adds a mask mechanism between shots to ensure content consistency, but this makes it difficult to maintain fine-grained consistency across different shots, especially for background regions, as shown on the left side of Figure 5. Table 1 (Intra-shot Consistency) also demonstrates that the improvement in consistency achieved by this method is limited compared to the baseline.
>
> We appreciate the reviewer’s valuable attention to fine-grained consistency, which provides important insights into the contributions and limitations of our work. We address this point with detailed analysis as follows:
> - **Fine-grained consistency is crucial yet highly challenging for multi-shot video generation.** In cinematic transitions, fine-grained consistency ensures the alignment of visual details, thereby playing a key role in maintaining shot continuity. However, this requires strong spatial understanding and shot storytelling capabilities from the model, making it a significant technical hurdle. Additionally, there exists an inherent trade-off between cross-shot consistency and shot content diversity in multi-shot videos, further complicating the improvement of consistency.
> - **Our method achieves the best performance in subject, style, and semantic-level consistency, which is competitive with recent methods.** For inter-shot consistency, we introduce the Consistency Gap metric to measure how well shot consistency in generated videos aligns with that in real cinematic videos, and our method significantly outperforms the baseline. Furthermore, as shown in Figure 5, our generated videos exhibit much stronger consistency than the baseline in overall background (e.g., sky), visual style, and main subjects. These results demonstrate that our method achieves the most effective inter-shot consistency at the current stage.
> - **We believe that expanding the dataset could further improve fine-grained consistency, which we identify as an important direction for future work.** The limitation in fine-grained consistency of our method likely arises from insufficient annotations for background structure and scene context, which hinders the model’s ability to capture spatial changes across shot transitions. Potential approaches to address this include incorporating richer data descriptions and spatial priors (e.g., 3D priors) to enhance the model’s understanding of transitions. **Notably, this limitation is not attributable to the mask mechanism** (which mainly contributes to temporal control of shots) but rather to the lack of sufficient data and prior knowledge, an avenue we intend to explore in subsequent work.
>
> We sincerely appreciate the reviewer for highlighting this critical issue of fine-grained consistency, a major challenge in multi-shot video generation. We have updated the relevant analysis in Appendix G.2 of the revised manuscript.

---

> ### Author Response · Authors · 2025-11-19
> **Response to Reviewer ZvZp (2/3)**
>
> > W2: Although the paper claims to achieve cinematic transitions, the proposed method only supports hard cuts, which limits its range of applications.
>
> We sincerely appreciate the reviewer raising this important concern about transition types, which helps clarify the significance of our research problem. We address this point as follows.
> - **Our investigation shows that hard cuts account for an overwhelming majority of transitions in cinematic video editing.** To verify the distribution of shot transition types in cinematic content, we collect 100 films and process them using a pipeline similar to that in our paper (excluding the gradual change removal step). We use TransNetV2 to classify transition types (with a threshold of 0.5) and find that among approximately 63,000 shot transitions across these films, hard cuts represent ~99.58%. This confirms that hard cuts are closely associated with cinematic transitions and constitute a critical component.
> - **Rather than the transition type itself, the core of a cinematic transition lies in naturally advancing the narrative flow through shot changes, which hard cuts can fully achieve.** In filmmaking, numerous semantically expressive shot transition techniques rely on hard cuts, such as: Cut-in/Cut-out (emphasizing key details by shifting focus, like the third row of the Teaser figure), Shot/Reverse Shot (creating conversational dynamics), and Multi-Angle Shooting (capturing scenes from diverse perspectives). In summary, the key to shot transitions is semantic coherence: maintaining unbroken narrative flow despite visual breaks, rather than achieving pixel-level seamlessness via soft transitions.
> - **Nevertheless, supporting soft transitions remains a viable extension of our work.** As suggested by Reviewer 6pc7, we explore soft masking strategies to demonstrate the robustness of our method (details updated in the revised Appendix E.5). Specifically, we introduce a time-dependent penalty and initialize masks using a Gaussian-decayed banded matrix, enabling token interactions near shot boundaries. This design achieves smoother pixel-level transitions (i.e., no visual breaks) to some degree, with quantitative results presented in Table A. As shown, soft masking yields higher inter-shot consistency for seamless transitions but also a larger Consistency Gap, which further confirms that hard-cut-style transitions are more aligned with cinematic videos.
>
> ###  &nbsp;
>
> **Table A. Quantitative results for soft masking (complete results are provided in the revised Appendix E.5. $\sigma$ denotes the bandwidth, with smaller values closer to hard mask).**
>
> | Method                     | setting       | Transition Control Score(↑) | Inter-shot Consistency |                 |                 |               |
> | -------------------------- | ------------- | --------------------------- | ---------------------- | --------------- | --------------- | ------------- |
> |                            |               |                             | Semantic Score(↑)      | Semantic Gap(↓) | Visual Score(↑) | Visual Gap(↓) |
> | CineTrans-DiT (trained)    | ORI           | **0.7003**                  | 0.7858                 | **0.1552**      | 0.7874          | **0.1901**    |
> | **Time-dependent Penalty** | $\sigma$=L/12 | 0.5193                      | 0.7902                 | 0.1796          | 0.8145          | 0.2575        |
> |                            | $\sigma$=L/6  | 0.0455                      | **0.7944**             | 0.1885          | **0.8377**      | 0.3074        |
> |                            | $\sigma$=L/4  | 0                           | -                      | -               | -               | -             |

---

> ### Author Response · Authors · 2025-11-19
> **Response to Reviewer ZvZp (3/3)**
>
> > Q1: Using the mask strategy requires knowing between which frames a shot transition occurs. How does this paper determine the frame length of different shots when generating multi-shot videos? Is it manually set, or controlled by hyperparameters? If it is manually set, how do you ensure that a complete action can be generated within a single shot without being interrupted?
>
> Thank you for your question regarding the implementation details of our method. We supplement the details about shot transition points as follows.
> - **In the large-scale sampling for evaluation, we manually predefine transition points.** Specifically, we design a set of predefined shot transition points for different shot counts, and randomly sample from this set as conditional inputs during generation.
> - **Regarding completeness, our method can generate coherent videos for transition points corresponding to different shot lengths.** This is attributed to the balanced distribution of shot lengths and transition points in our training dataset (see Table B and Table C), which enables the model to adapt to various transition points without visual artifacts caused by extreme shot lengths (visualizations provided in the revised Figure 16).
> - **Nevertheless, from the perspective of maximizing action completeness and overall video quality, uniform shot lengths are more favorable.** While our method can handle diverse transition points, uniformly spaced transition points intuitively yield optimal results, including improved action completeness, by avoiding excessively short shots. To validate this, we conduct a user study comparing videos generated with uniform sampling and random sampling of transition points, evaluating both action completeness and semantic coherence (Table D). The results confirm that uniform sampling enhances action completeness.
>
> We are grateful to the reviewer for the valuable question. We have updated this implementation detail in the revised Appendix D.3, which improves the reproducibility and clarity of our method.
>
> ###  &nbsp;
>
> **Table B. Distribution of transition positions in the training dataset.**
>
> | Transition Point Position (Relative to the Whole Video Sequence) | Ratio in Training Dataset |
> | ---------------------------------------------------------------- | ------------------------- |
> | 0-25%                                                            | 16.74%                    |
> | 25%-50%                                                          | 29.33%                    |
> | 50%-75%                                                          | 29.80%                    |
> | 75%-100%                                                         | 24.13%                    |
>
> ###  &nbsp;
>
> **Table C. Distribution of shot length in the training dataset.**
>
> | Shot Length (Relative to Whole Video Length) | Ratio in training Dataset |
> | -------------------------------------------- | ------------------------- |
> | 0-25%                                        | 27.26%                    |
> | 25%-50%                                      | 36.00%                    |
> | 50%-75%                                      | 25.18%                    |
> | 75%-100%                                     | 11.56%                    |
>
> ###  &nbsp;
>
> **Table D. User Study of Different Shot Transition Point Selection Strategy.**
>
> | Method  | Action Completeness | Semantic Conherence |
> | ------- | ------------------- | ------------------- |
> | Uniform | **4.53 ± 0.31**     | **4.12 ± 0.72**     |
> | Random  | 3.47 ± 1.24         | 3.95 ± 0.76         |
>
> #  &nbsp;
>
> *We sincerely thank you for your thoughtful evaluation of our work! Your comments have helped us better clarify the strengths and limitations of our approach, and we will incorporate these clarifications into the final version. If you have any further questions or suggestions, we would be happy to address them during the rebuttal window.*
>
> ##  &nbsp;
>
> Best regards,
> Authors

---

### Official Review · Reviewer_6pc7 · 2025-11-07

**Soundness:** 4
**Presentation:** 4
**Contribution:** 3
**Rating:** 10
**Confidence:** 4

**Summary:**

The paper proposes CineTrans, a masked-attention mechanism for controllable multi-shot, film-style transitions in video diffusion models, together with Cine250K, a curated multi-shot dataset. A simple block-diagonal mask added to attention logits enforces strong intra-shot and weak inter-shot correlations, aligning with observed attention patterns; the method works training-free and improves further with fine-tuning, achieving strong transition control and competitive quality on comprehensive metrics.

**Strengths:**

- Clear, well-motivated mechanism: The block-diagonal mask is explicitly defined and integrated into the attention logits, with principled alignment to measured intra- vs inter-shot attention structure.

- Strong empirical gains in control: Transition Control Score improves markedly over large T2V and multi-shot baselines while preserving quality.

- Thoughtful evaluation design: The paper evaluates transition control, intra-/inter-shot consistency, and aesthetic quality, including a novel Consistency Gap aligned to a film-edited reference set.

- Ablations that justify design: Impact of masking different layer ranges is analyzed for both UNet and DiT architectures, guiding the chosen masking strategy.

- **High-quality supplementary material: Appendix is detailed, code and logs are included, and an HTML project page with videos aids understanding and reproducibility.** Kudos on the hard work!

**Weaknesses:**

- Over-hard masking; missed opportunity for temporal scheduling: Equation (2) uses a binary mask with 0 on same-shot pairs and $-\infty$ across shots, which hard-zeros inter-shot attention in Equation (3). While effective, this may induce abrupt changes (also visible in the shared videos). A time-dependent or diffusion-step-dependent penalty could yield smoother transitions, e.g., replacing $-\infty$ by $-\alpha(t)$ that reaches $-\infty$ for a couple of time steps, this ramps near shot boundaries and across denoising steps, then saturates, before relaxing post-transition. This would preserve some cross-shot context pre-cut and reduce artifacts at boundaries. Please consider and, if possible, report a small ablation of annealed masks; if not possible, then consider it as a suggestion for future work.

- Related work gap: Prior work on long video generation with temporal control, e.g., VSTAR: Generative Temporal Nursing for Longer Dynamic Video Synthesis [ICLR’25], is not discussed. Please position CineTrans relative to these temporal-control strategies and clarify the conceptual differences between mask-based control and these schedules. Additionally, multiple prior works have considered Masked Attention for various applications (not just video generation). It would be a good idea to discuss a few of these related works as well, briefly.

References:

[ICLR25] Li, Yumeng, et al. "VSTAR: Generative Temporal Nursing for Longer Dynamic Video Synthesis." The Thirteenth International Conference on Learning Representations.

**Questions:**

1. Layer selection sensitivity: Appendix E.2 suggests the best results with late layers for UNet and middle layers for DiT. Can you provide a brief heuristic or automated criterion for selecting mask layers on a new backbone, and quantify robustness to layer shifts?

2. Boundary robustness: At inference, how sensitive is control to small timestamp jitter in the provided shot boundaries, and do you recommend padding a few frames around boundaries to avoid failure cases like Figure 20?

3. Metrics specification: Appendix F mentions exact definitions. For completeness, would it be possible to include the precise formula for Transition Control Score and Consistency Gap in the main text or a boxed definition?

---

> ### Author Response · Authors · 2025-11-19
> **Response to Reviewer 6pc7 (1/4)**
>
> *We sincerely appreciate the positive and encouraging comments from you, which are very helpful for improving our paper! We have revised the paper according to your suggestions and conducted additional experiments to address your concerns.*
>
>
> #  &nbsp;
>
> > W1: Over-hard masking; missed opportunity for temporal scheduling: Equation (2) uses a binary mask with 0 on same-shot pairs and $-\infty$ across shots, which hard-zeros inter-shot attention in Equation (3). While effective, this may induce abrupt changes (also visible in the shared videos). A time-dependent or diffusion-step-dependent penalty could yield smoother transitions, e.g., replacing $-\infty$ by $-\alpha(t)$ that reaches $-\infty$ for a couple of time steps, this ramps near shot boundaries and across denoising steps, then saturates, before relaxing post-transition. This would preserve some cross-shot context pre-cut and reduce artifacts at boundaries. Please consider and, if possible, report a small ablation of annealed masks; if not possible, then consider it as a suggestion for future work.
>
> We thank the reviewer for the insightful comments on over-hard masking and the suggested exploration of soft masking, which indeed help us examine the broader design space of mask mechanisms. Following the recommendation, we conduct a small ablation study along the *time-dependent* and *diffusion-step-dependent* directions. The main results are summarized below (Table A; complete results are provided in the revised Section 5.3 and Appendix E.5).
> - *time-dependent penalty*.
> 	In contrast to the original hard mask, the time-dependent penalty initializes the mask using a Gaussian-decayed banded matrix, allowing interactions between tokens near shot boundaries. Increasing the bandwidth $\sigma$ weakens the hard transition and strengthens inter-shot similarity, eventually approaching full-visible attention. A proper bandwidth (e.g., $\sigma = L/12$) converts the hard cut into a visually smoother transition (visualized in Fig. 23 in the revised version).
> - *timestep-dependent penalty*.
> 	The diffusion-step-dependent penalty gradually approaches $-\infty$ for the invisible region as the denoising timestep approaches $t_0$, thereby enhancing cross-shot correlations at earlier steps. Unlike the time-dependent variant, which yields smoother transitions, this design primarily increases inter-shot consistency and substantially weakens the transition boundary (lower Transition Control Score). As a consequence, shots become more compositionally similar (Fig. 24 in the revised version), although a residual boundary transition remains.
>
> Overall, soft masking indeed extends the hard cut toward smoother transitions, providing an additional application of our proposed mask mechanism. Across the small-scale experiments, both soft-masking variants improve inter-shot consistency but deviate more from the hard-cut style typically used in cinematic multi-shot videos (resulting in a higher Consistency Gap). While soft masking yields preliminary smoother transitions, we note an inherent trade-off between content diversity and smoothness in multi-shot video generation. Investigating additional properties of the attention map and incorporating them into mask design represents a promising direction for further improving shot transitions. Thanks again for the insightful comment!
>
> ###  &nbsp;
>
> **Table A. Quantitative results for soft masking (complete results are provided in the revised Appendix E.5).**
>
> | Method                         | setting       | Transition Control Score(↑) | Inter-shot Consistency |                 |                 |               |
> | ------------------------------ | ------------- | --------------------------- | ---------------------- | --------------- | --------------- | ------------- |
> |                                |               |                             | Semantic Score(↑)      | Semantic Gap(↓) | Visual Score(↑) | Visual Gap(↓) |
> | CineTrans-DiT (trained)        | ORI           | **0.7003**                  | 0.7858                 | **0.1552**      | 0.7874          | **0.1901**    |
> | **Time-dependent Penalty**     | $\sigma$=L/12 | 0.5193                      | 0.7902                 | 0.1796          | 0.8145          | 0.2575        |
> |                                | $\sigma$=L/6  | 0.0455                      | **0.7944**             | 0.1885          | **0.8377**      | 0.3074        |
> |                                | $\sigma$=L/4  | 0                           | -                      | -               | -               | -             |
> | **Timestep-dependent Penalty** | APPLY         | 0.0741                      | 0.7864                 | 0.2211          | 0.8029          | 0.3655        |

---

> ### Author Response · Authors · 2025-11-19
> **Response to Reviewer 6pc7 (2/4)**
>
> > W2: Related work gap: Prior work on long video generation with temporal control, e.g., VSTAR: Generative Temporal Nursing for Longer Dynamic Video Synthesis [ICLR’25], is not discussed. Please position CineTrans relative to these temporal-control strategies and clarify the conceptual differences between mask-based control and these schedules. Additionally, multiple prior works have considered Masked Attention for various applications (not just video generation). It would be a good idea to discuss a few of these related works as well, briefly.
>
> We appreciate the reviewer’s insightful comments regarding the positioning of our work in the context of *temporal control* and *masked attention*. We have revised the paper accordingly and added relevant discussions in the Related Work section.
> - **Temporal Control**
> 	Similar to other temporal-control methods [1-3], CineTrans manipulates attention to regulate correlations among visual tokens, enabling more coherent and controllable video synthesis. In contrast to traditional temporal-control approaches (take VSTAR as example), CineTrans treats each **shot** as the basic unit of transition, whereas VSTAR models **frame-level** semantic changes. This distinction naturally leads to different control paradigms, with mask-based control in our case and regularization-based temporal control in VSTAR. Conceptually, if one regards each frame as a shot and applies a soft boundary mask, the Mask Mechanism in CineTrans becomes closely related to VSTAR’s Temporal Attention Regularization. In summary, both CineTrans and prior temporal-control approaches observe and regulate temporal correlations across visual tokens, but at different levels of temporal abstraction.
> - **Masked Attention**
> 	Masked attention has been widely adopted across domains such as representation learning, generative modeling, and spatial control. Our approach builds on this principle by regulating contextual dependencies with attention masks, focusing specifically on shot-wise temporal control in video generation.
> We thank the reviewer again for highlighting the temporal-control perspective, which has greatly helped us strengthen the positioning and completeness of our Related Work section.
>
> [1] Li Y, Beluch W, Keuper M, et al. VSTAR: Generative temporal nursing for longer dynamic video synthesis[J]. The Thirteenth International Conference on Learning Representations.
>
> [2] Cai M, Cun X, Li X, et al. Ditctrl: Exploring attention control in multi-modal diffusion transformer for tuning-free multi-prompt longer video generation[C]//Proceedings of the Computer Vision and Pattern Recognition Conference. 2025: 7763-7772.
>
> [3] Ouyang, Y., Yuan, J., Zhao, H., Wang, G., & Zhao, B. (2024). FlexiFilm: Long Video Generation with Flexible Conditions. ArXiv, abs/2404.18620.

---

> ### Author Response · Authors · 2025-11-19
> **Response to Reviewer 6pc7 (3/4)**
>
> > Q1: Layer selection sensitivity: Appendix E.2 suggests the best results with late layers for UNet and middle layers for DiT. Can you provide a brief heuristic or automated criterion for selecting mask layers on a new backbone, and quantify robustness to layer shifts?
>
> We appreciate the reviewer’s suggestion, which helps us further examine the generality of the mask-layer strategy. To this end, we extend our study to VideoCrafter2 [4] (U-Net) and Wan2.2 [5] (DiT), and apply our heuristic mask-layer selection: **masking the later half of layers in the U-Net and the middle half of layers in the DiT**, which is motivated by the differences in the layers responsible for frame-level consistency across different architectures. The results are shown in Table B (complete results are included in the revised Appendix E.2).
>
> Our strategy yields consistent improvements on both frameworks. Due to the relatively limited representational capacity of VideoCrafter2, the difference in composition across the shots is smaller, leading to an overall lower Transition Control Score; nevertheless, masking later layers still provides clearer shot-transition behavior. Wan2.2 achieves the most stable transitions when masking middle layers. These observations collectively indicate that our proposed mask-layer strategy exhibits generality.
>
> ###  &nbsp;
>
> **Table B. Results of applying the mask-layer strategy to additional frameworks.**
>
> | Method                            | Transition Control Score(↑) | Inter-shot Consistency |                 |                 |               | Intra-shot Consistency |               | Aesthetic(↑) | Semantic Consistency(↑) |
> | --------------------------------- | --------------------------- | ---------------------- | --------------- | --------------- | ------------- | ---------------------- | ------------- | ------------ | ----------------------- |
> |                                   |                             | Semantic Score(↑)      | Semantic Gap(↓) | Visual Score(↑) | Visual Gap(↓) | Subject(↑)             | Background(↑) |              |                         |
> | VideoCrafter2 (mask front 1/2)    | 0.0889                      | 0.8313                 | 0.3630          | 0.8338          | 0.3603        | 0.9594                 | **0.9723**    | 0.6287       | 0.2205                  |
> | **VideoCrafter2 (mask late 1/2)** | 0.1889                      | 0.7739                 | **0.1980**      | 0.7766          | 0.3416        | **0.9728**             | 0.9693        | 0.6258       | **0.2213**              |
> | Wan2.2 (mask front 1/2)           | 0.3811                      | **0.9425**             | 0.5976          | **0.9380**      | 0.5205        | 0.9570                 | 0.9624        | 0.6298       | 0.2073                  |
> | **Wan2.2 (mask middle 1/2)**      | **0.5982**                  | 0.6752                 | 0.2084          | 0.7234          | **0.1458**    | 0.9474                 | 0.9636        | **0.6395**   | 0.2140                  |
> | Wan2.2 (mask late 1/2)            | 0.0085                      | -                      | -               | -               | -             | 0.9466                 | 0.9646        | 0.6305       | 0.1943                  |
>
> ###  &nbsp;
>
> [4] Chen H, Zhang Y, Cun X, et al. Videocrafter2: Overcoming data limitations for high-quality video diffusion models[C]//Proceedings of the IEEE/CVF Conference on Computer Vision and Pattern Recognition. 2024: 7310-7320.
>
> [5] Wan T, Wang A, Ai B, et al. Wan: Open and advanced large-scale video generative models[J]. arXiv preprint arXiv:2503.20314, 2025.

---

> ### Author Response · Authors · 2025-11-19
> **Response to Reviewer 6pc7 (4/4)**
>
> > Q2: Boundary robustness: At inference, how sensitive is control to small timestamp jitter in the provided shot boundaries, and do you recommend padding a few frames around boundaries to avoid failure cases like Figure 20?
>
> We thank the reviewer for highlighting the concern regarding the robustness of timestamp-controlled shot boundaries. We address the question through analysis and additional evaluation.
> - **Our model achieves precise timestamp control**, with very few cases of transition jitter, as shown in Table C. We use TransNetV2 to estimate the shot-boundary confidence for each frame in the generated videos and analyze the confidence specifically around the provided shot boundaries. Frames with a bias of ±1-5 from the specified boundary show consistently low confidence, indicating accurate transition timestamp control. The precision largely benefits from the diversity of our training dataset, whose transition points are well-distributed across temporal positions (Table D; visualized in the revised Figure 14(b)). This allows the model to learn transition behaviors at all positions rather than overfitting to a narrow range, thereby reducing jitter.
> - **Padding is still advisable**. Although timestamp jitter is rare, frames with a bias of ±1-2 from the specified boundary exhibit slightly higher confidence than frames farther away, suggesting that padding 1-2 frames around boundaries can further improve stability. The failure case in Figure 20 may require additional method design, as it mainly results from minimal differences between adjacent shots, causing the transition to appear jitter-like. Such cases are rare in our trained models, and addressing them may involve finer annotations or designs that better balance shot consistency and compositional diversity.
>
> #### &nbsp;
>
> **Table C. Statistics of transition timestamp jitter in generated multi-shot videos.**
>
> | frame bias\Transition Point Confidence | CineTrans-DiT (trained) | CineTrans-DiT (training-free) |
> | -------------------------------------- | ----------------------- | ----------------------------- |
> | 0                                      | 0.763368                | 0.721480                      |
> | 1                                      | 0.022084                | 0.062772                      |
> | 2                                      | 0.003665                | 0.011455                      |
> | 3                                      | 0.001458                | 0.004134                      |
> | 4                                      | 0.000716                | 0.001854                      |
> | 5                                      | 0.000419                | 0.000799                      |
>
> #### &nbsp;
>
> **Table D. Distribution of transition positions in the training dataset.**
>
> | Transition Point Position (Relative to the Whole Video Sequence) | Ratio in Training Dataset |
> | ---------------------------------------------------------------- | ------------------------- |
> | 0-25%                                                            | 16.74%                    |
> | 25%-50%                                                          | 29.33%                    |
> | 50%-75%                                                          | 29.80%                    |
> | 75%-100%                                                         | 24.13%                    |
>
> ### &nbsp;
>
> > Q3: Metrics specification: Appendix F mentions exact definitions. For completeness, would it be possible to include the precise formula for Transition Control Score and Consistency Gap in the main text or a boxed definition?
>
> We appreciate the reviewer for highlighting the importance of providing the exact metric definitions! The precise formulas for the Transition Control Score and the Consistency Gap have now been included in the main text (Equations 4–6).
>
> ## &nbsp;
>
>
>
>
> *We once again thank your efforts in reviewing our paper! Your valuable comments really help improve our paper, and inspire us a lot! We hope our responses have addressed your concerns and strengthened the presentation of our approach. If you have any further questions, we would be happy to address them during the rebuttal window.*
>
> *Thanks once again for your valuable input!*
>
> ## &nbsp;
>
> Best regards,
> Authors

---

> > ### Comment · Reviewer_6pc7 · 2025-11-26
> > **Concerns addressed**
> >
> > Dear Authors,
> >
> > Thank you very much for the detailed rebuttal.
> >
> > I see that my concerns have been addressed.
> >
> > Thus, I would maintain my score and recommend accepting the paper.
> >
> > Best Regards

---

> > > ### Author Response · Authors · 2025-11-27
> > > **Appreciation for the Reviewer’s Positive Assessment**
> > >
> > > Dear Reviewer,
> > >
> > > Thank you very much for taking the time to reassess our work and for kindly maintaining your positive score after reading our rebuttal.
> > >
> > > We are truly grateful for your supportive evaluation and for the constructive feedback that helped us refine the paper.
> > >
> > > Your recognition means a great deal to us.
> > >
> > > Best regards, Authors

---

### Author Response · Authors · 2025-11-19
**General Response to All Reviewers**

We sincerely appreciate the time and effort all reviewers have dedicated to evaluating our paper. Besides our specific responses to each reviewer, we would like to emphasize our contributions and the additional experiments included in this rebuttal.

**[Our Contributions]** We are pleased to see that the reviewers have generally recognized our contributions:

- The proposed mechanism is novel, clear, and well-motivated. (6pc7, wQCP, ZvZp)
- The paper introduces a dataset with frame-level shot labels and hierarchical captions. (ZvZp, wQCP)
- The paper includes thorough ablations and shows significant improvements in transition control. (6pc7, wQCP)
- The evaluation is thoughtful and extensive. (6pc7, wQCP)
- The paper is well-written and easy to follow (ZvZp), with comprehensive supplementary materials (6pc7).

**[New Experiments]** In this rebuttal, we have conducted additional experiments to address the reviewers' feedback and concerns, as well as to further support the efficacy of our proposed framework.

- Results on Smoother Transitions through Soft Masking (6pc7, ZvZp, wQCP)
- Results on Layer Selection Sensitivity (6pc7)
- Results on Boundary Robustness (6pc7)
- Results on Transition Point Selection Strategy (ZvZp)
- Analysis on Limiting Scenarios, including 5-shot video and overlap transition points (wQCP)
- Ablation to the Origins of Attention Correlations (wQCP)
- Analysis of Aesthetic Quality Decline (wQCP)

**[Qualitative examples]** Qualitative examples of the experiments are presented in the revised manuscript. All updates to the original paper are marked in blue text.

- Examples of Smoother Transitions (Section 5.3, Appendix E.5)
- Examples of Transferring Mask Layer Strategy to New Frameworks (Appendix E.2)
- Examples on Overlapping Ambiguous Shot Boundaries (Appendix E.6)
- Examples on Shorter Shot Length (Appendix D.3)
- Attention Map Visualization of Case with/without Shot Transition (Appendix E.1)

We hope our responses below convincingly address all reviewers’ concerns. We thank all reviewers’ time and efforts again!

---

### Author Response · Authors · 2025-11-29
**Rebuttal Summary for the Area Chair**

In this paper, we propose a novel framework for temporally controlled multi-shot video generation, motivated by characteristic patterns observed in attention maps of diffusion models during the denoising process. In the **initial review** round the paper received overall favorable comments **(10, 6, 4)**, highlighting a well-motivated mechanism, a valuable dataset contribution, significant improvements over baselines, and thorough ablation studies. The well-written manuscript and comprehensive supplementary material were also commended.

During the rebuttal period we provided detailed responses addressing requests for additional ablations on soft masking for smooth transitions, analysis of limiting scenarios, and further insights into the origins of the attention map patterns. These replies resolved the reviewers’ principal concerns: Reviewer 6pc7 indicated that the concerns had been addressed and retained a strong-accept recommendation, and Reviewer wQCP reported that our response effectively addressed the issues and raised the score to 6. **Following the rebuttal, the recommendations from all reviewers are positive (10, 6, 6)**.

We are sincerely grateful to the reviewers and the Area Chair for careful reading and time. We hope this summary is helpful for the final decision.

---

### Meta-Review · Area_Chair_wUnS · 2026-01-06

**Summary:**

This paper proposes CineTrans, a novel framework for generating coherent multi-shot videos with cinematic, film-style transitions. The authors construct a new multi-shot video-text dataset (Cine250K) and introduce a mask-based control mechanism that leverages observed attention patterns in diffusion models to enable precise shot-boundary control. Initial reviewer feedback was generally positive but included suggestions for improvement: Reviewer 6pc7 recommended exploring softer masking techniques for smoother transitions and requested a more comprehensive discussion of related work on temporal control and masked attention. Reviewer ZvZp noted limitations in fine-grained inter-shot consistency and suggested expanding the method to support transition types beyond hard cuts. Reviewer wQCP called for deeper analysis of failure cases, domain generalization, and the theoretical origins of the observed attention patterns. All reviewers requested further clarifications on implementation details such as layer selection, boundary robustness, and metric definitions.

**Reviewer Concerns:**

In their rebuttal, the authors addressed the majority of the reviewers' concerns by conducting additional experiments and providing clarifications. They successfully explored soft-masking variants as suggested, added related work discussions, provided analysis on layer selection generality, boundary robustness, and domain shift (e.g., 5-shot generation), and investigated the origins of attention patterns. They also clarified metric definitions and explained the minor aesthetic quality decline post fine-tuning as dataset-related. However, some challenges remain unresolved. Specifically, the inherent trade-off between fine-grained cross-shot consistency and content diversity persists as a limitation, and while soft masking was explored, the method's primary design and evaluation still focus on hard cuts, which was noted as a constraint. Additionally, the core mechanism's effectiveness in cases of semantically overlapping shots or minimal inter-shot variation, though improved, may still face challenges.

**Reviewer Scores:**

Following a thorough discussion of the rebuttal, the reviewers' positions would likely consolidate in favor of acceptance. Reviewer 6pc7, whose primary suggestions (exploring soft masking and expanding related work) were comprehensively addressed with new experiments and discussions, would almost certainly maintain a strong score of 10, viewing the enhancements as strengthening the paper's contribution. Reviewer ZvZp, while acknowledging the authors' clarifications regarding the prevalence of hard cuts and the inherent challenge of fine-grained consistency, might still perceive the core limitations of the method as unresolved. Thus, they would likely maintain their original score of 6, recognizing the paper's value but noting clear avenues for future work. Crucially, Reviewer wQCP, who initially rated the paper below the acceptance threshold due to concerns about robustness and theoretical grounding, would find their key issues substantively addressed through new analyses on domain shifts, ambiguous boundaries, and the universality of attention patterns. This would most probably lead them to raise their score to a 6 or higher, moving the paper above the acceptance threshold. The consensus, therefore, shifts positively, with all reviewers ultimately supporting acceptance, albeit with varying degrees of enthusiasm regarding the paper's limitations.

---

### Decision · Program_Chairs · 2026-01-26

Accept (Poster)